# RELATIONAL GRAPH TRANSFORMER

**Vijay Prakash Dwivedi**[1]    **Sri Jaladi**[1]    **Yangyi Shen**[1]    **Federico López**[2]
**Charilaos I. Kanatsoulis**[1]    **Rishi Puri**[3]    **Matthias Fey**[2]    **Jure Leskovec**[1,2]
[1]Stanford University,    [2]Kumo.AI,    [3]NVIDIA
{vdwivedi,jure}@cs.stanford.edu

## ABSTRACT

Relational Deep Learning (RDL) is a promising approach for building state-of-the-art predictive models on multi-table relational data by representing it as a heterogeneous temporal graph. However, commonly used Graph Neural Network models suffer from fundamental limitations in capturing complex structural patterns and long-range dependencies that are inherent in relational data. While Graph Transformers have emerged as powerful alternatives to GNNs on general graphs, applying them to relational entity graphs presents unique challenges: (i) Traditional positional encodings fail to generalize to massive, heterogeneous graphs; (ii) existing architectures cannot model the temporal dynamics and schema constraints of relational data; (iii) existing tokenization schemes lose critical structural information. Here we introduce the Relational Graph Transformer (RELGT), the first graph transformer architecture designed specifically for relational tables. RELGT employs a novel multi-element tokenization strategy that decomposes each node into five components (features, type, hop distance, time, and local structure), enabling efficient encoding of heterogeneity, temporality, and topology without expensive precomputation. Our architecture combines local attention over sampled subgraphs with global attention to learnable centroids, incorporating both local and database-wide representations. Across 21 tasks from the RelBench benchmark, RELGT consistently matches or outperforms GNN baselines by up to 18%, establishing Graph Transformers as a powerful architecture for Relational Deep Learning[1].

## 1 INTRODUCTION

Real-world enterprise data, such as financial transactions, supply chain data, e-commerce records, product catalogs, customer interactions, and electronic health records, are predominantly stored in relational databases (Codd, 1970). These databases typically consist of multiple tables, each dedicated to different entity types, interconnected through primary-foreign key links. This abstraction underpins large quantities of complex, dynamically updated data that scale with business volume, storing potentially immense, unexploited knowledge (Fey et al., 2024). However, extracting predictive patterns from such data has traditionally depended on manual feature engineering within complex machine learning pipelines, requiring the transformation of multi-table records into flat feature vectors suitable for models like deep neural networks and decision trees (Chen & Guestrin, 2016).

**Relational Deep Learning.** To enable end-to-end deep learning, relational databases can be represented as relational entity graphs (Fey et al., 2024), where nodes correspond to entities and edges capture primary-foreign key relationships. This graph-based representation allows Graph Neural Networks (GNNs) to learn abstract features directly from the underlying data structure, effectively modeling complex dependencies for various downstream prediction tasks. With this setup, which is termed as Relational Deep Learning (RDL), GNNs reduce or eliminate the need for manual feature engineering and often lead to better performance (Robinson et al., 2024), at a fraction of the traditional model development cost.

**Existing gaps.** Despite their effectiveness, standard message-passing GNN architectures (Gilmer et al., 2017; Kipf & Welling, 2016; Hamilton et al., 2017; Velickovic et al., 2017) have notable limitations, such as insufficient structural expressiveness (Xu et al., 2019; Morris et al., 2019; Loukas,

---

[1]https://github.com/snap-stanford/relgt

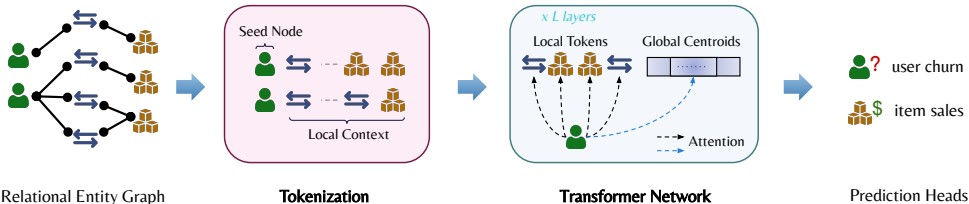

Figure 1: Overview of the RELGT architecture. First, the input relational entity graph (REG) is converted into tokens where each training seed node (such as the *customer* node in this example) gets a fixed number of neighboring nodes, which are encoded with a multi-element tokenization strategy. These tokens are then passed through a Transformer network that builds both local and global representations, which are then fed to downstream prediction layers.

2019) and restricted long-range modeling capabilities (Alon & Yahav, 2020). For example, consider an e-commerce database with three tables: *customers*, *transactions*, and *products*, which can be represented as a relational entity graph as in Figure 1. In a standard GNN, *transactions* are always two hops away from each other, connected only through shared customers. This creates an information bottleneck: *transaction*-to-*transaction* patterns require multiple layers of message passing, while *product* relationships remain entirely indirect in shallow networks. Furthermore, *products* would never directly interact in a two-layer GNN (Robinson et al., 2024), as their messages must pass through both a transaction and a customer, highlighting the inherent structural constraints of GNN architectures that restrict capturing long-range dependencies.

Graph Transformers (GTs) have emerged as more expressive models for graph learning, utilizing self-attention in the full graph to increase the range of information flow and additionally, incorporating positional and structural encodings (PEs/SEs) to better capture graph topology (Dwivedi & Bresson, 2021; Ying et al., 2021; Rampášek et al., 2022). These advances have produced strong results across domains (Müller et al., 2023), including foundation models for molecular graphs (Sypetkowski et al., 2024). However, many GT designs are limited to non-temporal, homogeneous, and small-scale graphs, assumptions that do not hold for relational entity graphs (REGs) (Fey et al., 2024), which are typically (i) heterogeneous, with different tables representing distinct node types; (ii) temporal, with entities often associated with timestamps and requiring careful handling to prevent data leakage; (iii) large-scale, containing millions or more records across multiple interconnected tables. In particular, existing PEs often require precomputation, depend on graph size, and typically do not scale well to large, heterogeneous, or dynamic graphs (Cantürk et al., 2023; Kanatsoulis et al., 2025). For instance, node2vec (Grover & Leskovec, 2016), while more efficient than Laplacian or random walk PEs, can become prohibitively expensive and impractical to compute on massive graphs (Postăvaru et al., 2020). These limitations, along with the inability to capture the multi-dimensional complexity of relational structures, render current GTs inadequate for relational databases.

**Present work.** We introduce the **Relational Graph Transformer (RELGT)**, the first Graph Transformer specifically designed for relational entity graphs. RELGT addresses key gaps in existing methods by enabling effective graph representation learning within the RDL framework. It is a unified model that explicitly captures the temporality, heterogeneity, and structural complexity inherent to relational graphs. We summarize the architecture as follows (Figure 1):

- **Tokenization:** We develop a multi-element tokenization scheme that converts each node into structurally enriched tokens. By sampling fixed-size subgraphs as local context windows and encoding each node's features, type, hop distance, time, and local structure, RELGT captures fine-grained graph properties without expensive precomputation at the subgraph or graph level.

- **Attention:** We develop a transformer network that combines local and global representations, adapting existing GT architectures (Rampášek et al., 2022). The model extracts features from the local tokens while simultaneously attending to learnable global tokens that act as soft centroids, effectively balancing fine-grained structural modeling with database-wide patterns.

- **Validation:** We showcase RELGT's effectiveness through a comprehensive evaluation on 21 tasks from RelBench (Robinson et al., 2024). RELGT consistently outperforms GNN baselines, with gains of up to 18%, establishing transformers as a powerful architecture for relational deep learning. Compared to HGT, a strong GT baseline for heterogeneous graphs, RELGT achieves better results without added computational cost, even when HGT uses Laplacian eigenvectors for PE.

## 2 BACKGROUND

### 2.1 RELATIONAL DEEP LEARNING

Relational Deep Learning is an end-to-end learning framework that converts relational databases into graph structures, enabling direct use of GNNs for representation learning (Fey et al., 2024).

**Definitions.** Formally, we can define a **relational database** as the tuple $(T, R)$ comprising a collection of tables $T = \{T_1, \ldots, T_n\}$ connected through inter-table relationships $R \subseteq T \times T$. A link $(T_{\text{fkey}}, T_{\text{pkey}}) \in R$ denotes a foreign key in one table referencing a primary key in another. Each table contains entities (rows) $\{v_1, \ldots, v_{n_T}\}$, with each entity typically consisting of: (1) a unique identifier (primary key), (2) references to other entities (foreign keys), (3) entity-specific attributes, and (4) timestamp information indicating when the entity was created or modified. The structure of relational databases inherently forms a graph representation, called as **relational entity graphs** (REGs). An REG is formally defined as a heterogeneous temporal graph $G = (V, E, \phi, \psi, \tau)$, where nodes $V$ represent entities from the database tables, edges $E$ represent primary-foreign key relationships, $\phi$ maps nodes to their respective types based on source tables, $\psi$ assigns relation types to edges, and $\tau$ captures the temporal dimension through timestamps (Fey et al., 2024).

**Challenges.** Relational entity graphs exhibit three distinctive properties that set them apart from conventional graph data. First, their structure is fundamentally schema-defined, with topology shaped by primary-foreign keys rather than arbitrary connections, creating specific patterns of information flow that require specialized modeling approaches. Second, they incorporate temporal dynamics, as relational databases track events and interactions over time, necessitating techniques like time-aware neighbor sampling to prevent future information from leaking into past predictions. Third, they display multi-type heterogeneity, as different tables correspond to different entity types with diverse attribute schemas and data modalities, presenting challenges in creating unified representations that effectively integrate information across diverse node and edge types (Schlichtkrull et al., 2018; Wang et al., 2019). These characteristics create both challenges and opportunities for GNN architectures, requiring models that can simultaneously address temporal evolution, heterogeneous information, and schema-constrained structures while processing potentially massive multi-table datasets.

### 2.2 RDL METHODS

The baseline GNN approach introduced by (Robinson et al., 2024) for RDL uses a heterogeneous GraphSAGE (Hamilton et al., 2017) model with temporal-aware neighbor sampling, which demonstrates significant improvements compared to traditional tabular methods like LightGBM (Ke et al., 2017) across most of the tasks in the RelBench benchmark. This baseline architecture leverages PyTorch Frame's multi-modal feature encoders (Hu et al., 2024) to transform diverse entity attributes into initial feature embeddings that serve as input to the GNN. Several specialized architectures have been developed to address specific challenges in relational entity graphs. RelGNN (Chen et al., 2025) introduces composite message-passing with atomic routes to facilitate direct information exchange between neighbors of bridge and hub nodes, commonly found in relational structures. Similarly, ContextGNN (Yuan et al., 2024) employs a hybrid approach, combining pair-wise and two-tower representations, specifically optimized for recommendation tasks in RelBench.

Beyond pure GNN approaches, retrieval-augmented generation techniques (Wydmuch et al., 2024) and hybrid tabular-GNN methods (Lachi et al., 2024) have also demonstrated comparable or superior performance to the standard GNN baseline, while showing the use of LLMs (Grattafiori et al., 2024) and inference speedups, respectively. These approaches confirm the effectiveness of graph, tabular, and LLM-based methods for downstream predictions in RDL. However, these methods typically optimize specific aspects of the problem, failing to incorporate broader advances from GTs in general.

### 2.3 GRAPH TRANSFORMERS

Graph Transformers extend the self-attention mechanism from sequence modeling (Vaswani et al., 2017) to graph-structured data, offering powerful alternatives to traditional GNNs (Dwivedi & Bresson, 2021). These models typically restrict attention to local neighborhoods, functioning as message-passing networks with attention-based aggregation (Joshi, 2020; Bronstein et al., 2021), while positional encodings are developed based on Laplacian eigenvectors (Dwivedi et al., 2020).

Subsequent Graph Transformers incorporate global attention mechanisms, allowing all nodes to attend to one another (Ying et al., 2021; Mialon et al., 2021; Kreuzer et al., 2021). This moves beyond the local neighborhood limitations of standard GNNs (Alon & Yahav, 2020), albeit at the cost of significantly increased computational complexity.

Modern GT architectures have improved the aforementioned early works by creating effective structural encodings and ensuring scalability to medium and large-scale graphs. For structural expressiveness of the node tokens, several positional and structural encoding methods have been developed (Dwivedi et al., 2022; Cantürk et al., 2023; Lim et al., 2022; Huang et al.; Kanatsoulis et al., 2025) to inject the input graph topology. For scalability, various strategies have emerged including hierarchical clustering that coarsens graphs (Zhang et al., 2022; Zhu et al., 2023), sparse attention mechanisms that reduce computational cost (Rampášek et al., 2022; Shirzad et al., 2023), and neighborhood sampling techniques for processing massive graphs (Zhao et al., 2021; Chen et al., 2022; Dwivedi et al., 2023). Models like GraphGPS (Rampášek et al., 2022) combine these advances through hybrid local-global designs that maintain Transformers' global context advantages while ensuring practical efficiency when scaling to medium and large graph datasets. However, these approaches exhibit several key limitations: they are largely confined to static graphs, and lack mechanisms to handle multiple node and edge types. While specialized Transformers for heterogeneous graphs exist (Hu et al., 2020; Mao et al., 2023; Zhu et al., 2023; Zhai et al., 2024), integrating them, alongside other aforementioned methods, into the RDL pipeline remains challenging. This is primarily because adapting PEs under precomputation constraints is difficult, compounded by the complexity of modeling large-scale, temporal, and heterogeneous relational entity graphs (REGs).

### 2.4 ADDRESSING CHALLENGES

While heterogeneous graph transformers (Hu et al., 2020) and temporal graph methods exist, no prior GT architecture effectively handles relational entity graphs where heterogeneity, temporality, and rich entity attributes co-occur within schema-defined database structures. Heterogeneous Temporal Graph Transformers like HTGformer (Wang, 2025) process heterogeneity and temporality through separate, iterative modules without graph positional encodings, a component now considered essential in modern GTs (Rampášek et al., 2022; Müller et al., 2023), and do not address the multimodal attributes or schema constraints inherent to relational databases (Hu et al., 2024). Existing subgraph-based GTs (Zhao et al., 2021; Chen et al., 2022) focus on scalability for homogeneous graphs without mechanisms for heterogeneity or temporal dynamics.

We address these limitations by recognizing that relational entity graphs require a rethinking on how their multimodal attributes and comprehensive graph structure are jointly processed. We systematically decompose the information coming from the REGs into specialized components that can be independently learned and composed, as we describe in detail in Section 3. This principled design enables GTs to handle the unique combination of heterogeneity, temporality, and schema-defined structures in relational databases without expensive global precomputation.

## 3 RELGT: RELATIONAL GRAPH TRANSFORMER

### 3.1 TOKENIZATION

Traditional Transformers in NLP represent text through tokens with two primary elements: (i) **token identifiers** (or features) that denotes the token from a vocabulary set and (ii) **positional encodings** that represent sequential structure (Vaswani et al., 2017). For example, a token can correspond to a word and its positional encoding can correspond to its order in the input sentence. Similarly, Graph Transformers *generally* adapt this two-element representation to graphs, where nodes are tokens with features, and graph positional encodings provide structural information. Although this two-element approach works well for homogeneous static graphs, it becomes computationally inefficient when trying to encode multiple aspects of graph structural information for REGs.

In particular, capturing heterogeneity, temporality, and schema-defined structure (as defined in Section 2.1) through a single positional encoding scheme would either require complex, multi-stage encoding or result in significant information loss about the rich relational context. For instance, if we were to extend existing PEs for REGs, several practical challenges emerge: (i) standard Laplacian or random walk-based PEs would need significant modification to differentiate between multiple

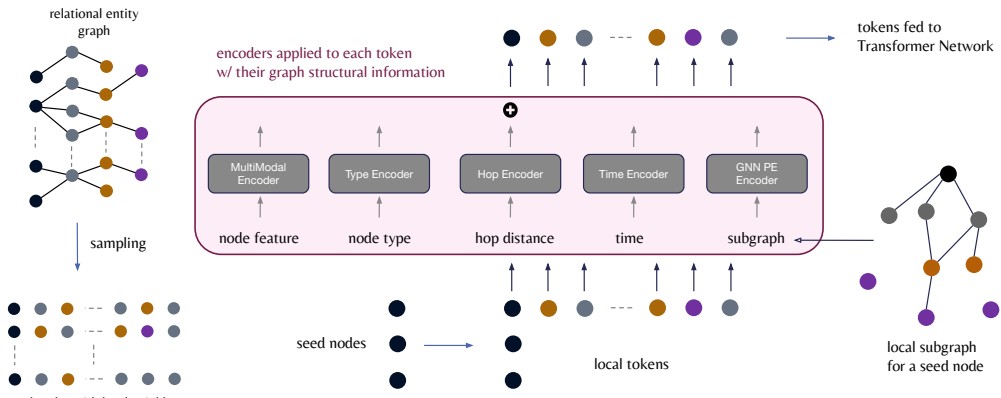

Figure 2: The tokenization procedure. A temporal-aware subgraph sampling step extracts a fixed set of local tokens for each training seed node, denoted by the node in black. Each token incorporates its respective graph structure information, which are element-wise transformed to a common embedding space and combined to form the effective token representation to be fed to the Transformer network.

node types (*e.g.*, customers vs. products vs. transactions), (ii) these encodings lack mechanisms to incorporate temporal dynamics critical for time-sensitive predictions (*e.g.*, capturing that a user's recent purchases are more relevant than older ones), and (iii) the scale of relational databases makes global PE computation in REGs prohibitively expensive. With millions of records across tables, precomputation would only be feasible on small subgraphs, resulting in incomplete structural context.

### 3.1.1 PROPOSED APPROACH

RELGT overcomes these limitations through a multi-element token representation approach, without any computational overhead concerning the dependency on the number of nodes in the input REG. Rather than trying to compress all structural information into a single positional encoding, we decompose the token representation into distinct elements that explicitly model different aspects of relational data. This decoupled design allows each component to capture a specific characteristic of REGs: node features represent entity attributes, node types encode table-based heterogeneity, hop distance preserves relative distances among nodes in a local context, time encodings capture temporal dynamics, and GNN-based positional encodings preserve local graph structure.

**Sampling and token elements.** The tokenization process in RELGT converts a REG $G = (V, E, \phi, \psi, \tau)$ into sets of tokens suitable for processing by the Transformer network. Specifically, as shown in Fig 2, for each training seed node $v_i \in V$, we first sample a fixed set of $K$ neighboring nodes $v_j$ from within 2 hops of the local neighborhood using temporal-aware sampling[2], ensuring that only nodes with timestamps $\tau(v_j) \leq \tau(v_i)$ are included to prevent temporal leakage. Each token in this set is represented by a **5-tuple**: $(x_{v_j}, \phi(v_j), p(v_i, v_j), \tau(v_j) - \tau(v_i), \text{GNN-PE}_{v_j})$, where, (i) node feature ($x_{v_j}$) denotes the raw features derived from entity attributes in the database, (ii) node type ($\phi(v_j)$) is a categorical identifier corresponding to the entity's originating table, (iii) relative hop distance ($p(v_i, v_j)$) captures the structural distance between the seed node $v_i$ and the neighbor node $v_j$, (iv) relative time ($\tau(v_j) - \tau(v_i)$) represents the temporal difference between the neighbor and seed node, and (v) finally, subgraph based PE (GNN-PE$_{v_j}$) provides a graph PE for each node within the sampled subgraph, generated by applying a lightweight GNN to the subgraph's adjacency matrix with random node feature initialization (Sato et al., 2021; Kanatsoulis et al., 2025).

**Encoders.** Each element in the 5-tuple is processed by a specialized encoder before being combined into the final token representation, as illustrated in Figure 2.

*1. **Node Feature Encoder.*** The node features $x_{v_j}$, representing the columnar attributes of the node $v_j$ in REG (which corresponds to a table row in a database), are encoded into a $d$-dimensional embedding. Each modality, such as numerical, categorical, multi-categorical, text, and image data, is encoded separately using modality-specific encoders following (Hu et al., 2024), and the resulting

---

[2]When fewer than $K$ neighbors are available within 2 hops, we use randomly selected nodes as fallback tokens to maintain the fixed size $K$, ensuring consistent computational complexity regardless of local structure.

representations are then aggregated into a unified $d$-dimensional embedding.

$$h_{\text{feat}}(v_j) = \text{MultiModalEncoder}(x_{v_j}) \in \mathbb{R}^d \tag{1}$$

where MultiModalEncoder($\cdot$) is unified feature encoder adapted from (Hu et al., 2024).

*2. **Node Type Encoder.*** The node type encoding steps converts each table-specific entity type $\phi(v_j)$ to a $d$-dimensional representation, incorporating the heterogeneous information from the input data.

$$h_{\text{type}}(v_j) = W_{\text{type}} \cdot \text{onehot}(\phi(v_j)) \in \mathbb{R}^d \tag{2}$$

where $\phi(v_j)$ is the node type of $v_j$, $W_{\text{type}} \in \mathbb{R}^{d \times |T|}$ is the learnable weight matrix, $|T|$ is the number of node types, and onehot($\cdot$) is the one-hot encoding function.

*3. **Hop Encoder.*** The relative hop distance $p(v_i, v_j)$, that captures the structural proximity between the seed node $v_i$ and a neighbor node $v_j$, is encoded into a $d$-dimensional embedding as:

$$h_{\text{hop}}(v_i, v_j) = W_{\text{hop}} \cdot \text{onehot}(p(v_i, v_j)) \in \mathbb{R}^d \tag{3}$$

with $p(v_i, v_j)$ being the relative hop distance between seed node $v_i$ and neighbor node $v_j$, and $W_{\text{hop}} \in \mathbb{R}^{d \times h_{\max}}$ the learnable matrix mapping hop distances (up to $h_{\max}$).

*4. **Time Encoder.*** The time encoder linearly transforms the time difference $\tau(v_j) - \tau(v_i)$ between a neighbor node $v_j$ and the seed node $v_i$:

$$h_{\text{time}}(v_i, v_j) = W_{\text{time}} \cdot (\tau(v_j) - \tau(v_i)) \in \mathbb{R}^d \tag{4}$$

where $\tau(v_j) - \tau(v_i)$ is the relative time difference, and $W_{\text{time}} \in \mathbb{R}^{d \times 1}$ are learnable parameters.

*5. **Subgraph PE Encoder.*** Finally, for capturing local graph structure that can otherwise not be represented by other token elements, we apply a light-weight GNN to the subgraph. This GNN encoder effectively preserves important structural relationships, such as complex cycles and quasi-cliques between entities (Kanatsoulis & Ribeiro, 2024), as well as parent-child relationships (e.g., a *product* node within the local subgraph corresponding to specific *transactions*), and can be written as:

$$h_{\text{pe}}(v_j) = \text{GNN}(A_{\text{local}}, Z_{\text{random}})_j \in \mathbb{R}^d \tag{5}$$

where $\text{GNN}(\cdot, \cdot)_j$ is a light-weight GNN applied to the local subgraph yielding the encoding for node $v_j$, $A_{\text{local}} \in \mathbb{R}^{K \times K}$ is the adjacency matrix of the sampled subgraph of $K$ nodes, and $Z_{\text{random}} \in \mathbb{R}^{K \times d_{\text{init}}}$ are randomly initialized node features for the GNN (with $d_{\text{init}}$ as the initial feature dimension).

One key advantage of using random node features in this GNN encoder is that it breaks structural symmetries between the subgraph topology and node attributes, thereby increasing the expressiveness of GNN layers (Sato et al., 2021). However, a fixed random initialization would destroy permutation equivariance, a critical property for generalization. To address this, we resample $Z_{\text{random}}$ independently at every training step. This 'stochastic initialization' approach can be viewed as a relaxed version of the learnable PE method described in Kanatsoulis et al. (2025), thus approximately preserving permutation equivariance while retaining the expressivity gains afforded by the randomization.

At last, the effective token representation is formed by combining all encoded elements:

$$h_{\text{token}}(v_j) = O \cdot [h_{\text{feat}}(v_j) \,||\, h_{\text{type}}(v_j) \,||\, h_{\text{hop}}(v_i, v_j) \,||\, h_{\text{time}}(v_i, v_j) \,||\, h_{\text{pe}}(v_j)] \tag{6}$$

where $||$ denotes the concatenation of the individual encoder outputs, and $O \in \mathbb{R}^{5d \times d}$ is a learnable matrix to mix the embeddings. This multi-element approach provides a comprehensive token representation that explicitly captures node features, type information, structural position, temporal dynamics, and local topology without requiring expensive computation on the graph structure.

## 3.2 Transformer Network

The Transformer network in RELGT, shown in Fig. 3, processes the tokenized relational entity graph using a combination of local and global attention mechanisms, following the successful designs used in modern GTs (Rampášek et al., 2022; Wu et al., 2023; Kong et al., 2023; Dwivedi et al., 2023).

**Local module.** The local attention mechanism allows each seed node to attend to its $K$ local tokens selected during tokenization, capturing the fine-grained relationships defined by the database schema.

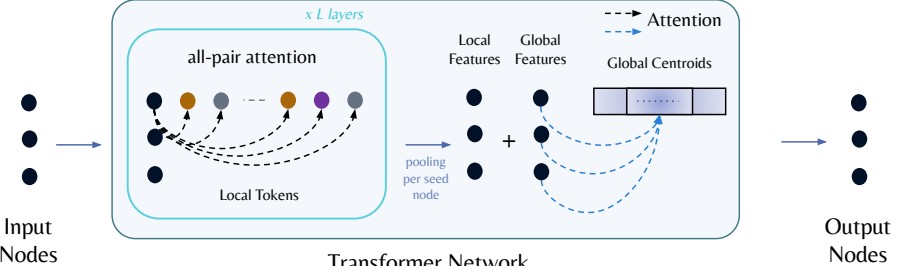

Figure 3: The Transformer network which processes the input tokens by first building local representations using the local tokens, then incorporating global context by attending to centroids that are dynamically updated during training. The final node representations combine both local structural details and global database context, enabling effective prediction across downstream tasks.

This mechanism is different from a GNN used in RDL (Robinson et al., 2024) in two key aspects: self-attention is used as the message-passing scheme and the attention is all-pair, *i.e.*, all nodes in the local $K$ set attend to each other. This is implemented using an $L$ layer Transformer (Vaswani et al., 2017) and provides a broader structural coverage compared to a baseline GNN (Robinson et al., 2024). A practical application of this improvement can be seen in the e-commerce example introduced in Sec. 1, where the proposed full-attention mechanism can directly connect seemingly unrelated products by identifying relationships through shared transactions or customer behaviors. This capability enables the model to capture subtle associations, such as customers frequently purchasing unexpected combinations of items. The local node representation $h_{\text{local}}(v_i)$ is obtained as:

$$h_{\text{local}}(v_i) = \text{Pool}(\text{FFN}(\text{Attention}(v_i, \{v_j\}_{j=1}^{K}))_L) \tag{7}$$

where, $L$ denotes the layers, FFN and Attention are standard components in a Transformer (Vaswani et al., 2017), and Pool denotes the aggregation of $\{v_j\}_{j=1}^{K}$ and $v_i$ using a learnable linear combination.

**Global module.** The global attention mechanism enables each seed node to attend to a set of $B$ global tokens representing centroids of all nodes in the graph, conceptually and is adapted from prior works (Kong et al., 2023; Dwivedi et al., 2023). These centroids are updated during training using an Exponential Moving Average (EMA) K-Means algorithm applied to seed node features in each mini-batch, providing a broader contextual view beyond the local neighborhood. The global representation is formulated as:

$$h_{\text{global}}(v_i) = \text{Attention}(v_i, \{c_b\}_{b=1}^{B}) \tag{8}$$

The final output representation of each node $v_i$ is obtained by combining local and global embeddings:

$$h_{\text{output}}(v_i) = \text{FFN}([h_{\text{local}}(v_i) \,||\, h_{\text{global}}(v_i)]) \tag{9}$$

with FFN being a feed forward network. The components of the Transformer in all stages follow standard instantiations with normalization and residual connections.

For downstream prediction, the combined representation of the seed node is passed through a task-specific prediction head. The model is trained end-to-end using suitable task specific loss functions. By leveraging multi-element token representations within a hybrid local-global Transformer architecture, RELGT effectively addresses the challenges of heterogeneity, temporal dynamics, and schema-defined structures inherent in relational entity graphs.

## 4 EXPERIMENTS

RELGT is evaluated on the recently introduced RDL Benchmark (RelBench) (Robinson et al., 2024). RelBench consists of 7 datasets from diverse relational database domains, including e-commerce, clinical records, social networks, and sports, among others. These datasets are curated from their respective source domains and consist a wide range of sizes, from 1.3K to 5.4M records in the training set for the prediction tasks, with a total of 47M training records. For each dataset, multiple predictive tasks are defined, such as predicting a user's engagement with an advertisement within the next four

Table 1: Test set results on the entity regression and classification tasks in RelBench. Best values are in **bold**. RDL: HeteroGNN baseline (Robinson et al., 2024), HGT: Heterogeneous GT (Hu et al., 2020), PE: Laplacian PE. Relative gains are expressed as percentage improvement over RDL baseline.

(a) MAE for entity regression. Lower is better

| Dataset | Task | RDL | HGT | HGT +PE | RelGT (ours) | % Rel. Gain |
|---------|------|-----|-----|---------|--------------|-------------|
| rel-f1 | driver-position | 4.022 | 4.2263 | 4.3921 | **3.9170** | 2.61 |
| rel-avito | ad-ctr | 0.041 | 0.0462 | 0.0483 | **0.0345** | 15.85 |
| rel-event | user-attendance | 0.258 | 0.2635 | 0.2611 | **0.2502** | 2.79 |
| rel-trial | study-adverse | 44.473 | 45.1692 | **42.6484** | 43.9923 | 1.08 |
| | site-success | 0.400 | 0.4428 | 0.4396 | **0.3263** | 18.43 |
| rel-amazon | user-ltv | 14.313 | 15.4120 | 15.8643 | **14.2665** | 0.32 |
| | item-ltv | 50.053 | 55.8683 | 55.8493 | **48.9222** | 2.26 |
| rel-stack | post-votes | **0.065** | 0.0679 | 0.0680 | **0.0654** | -0.62 |
| rel-hm | item-sales | 0.056 | 0.0641 | 0.0639 | **0.0536** | 4.29 |

(b) AUC for entity classification. Higher is better.

| Dataset | Task | RDL | HGT | HGT +PE | RelGT (ours) | % Rel. Gain |
|---------|------|-----|-----|---------|--------------|-------------|
| rel-f1 | driver-dnf | 0.7262 | 0.7077 | 0.7117 | **0.7587** | 4.48 |
| | driver-top3 | 0.7554 | 0.7075 | 0.7627 | **0.8352** | 10.56 |
| rel-avito | user-clicks | 0.6590 | 0.6376 | 0.6457 | **0.6830** | 3.64 |
| | user-visits | 0.6620 | 0.6432 | 0.6495 | **0.6678** | 0.88 |
| rel-event | user-repeat | **0.7689** | 0.6496 | 0.6536 | 0.7609 | -1.04 |
| | user-ignore | 0.8162 | **0.8247** | 0.8161 | 0.8157 | -0.06 |
| rel-trial | study-outcome | 0.6860 | 0.5837 | 0.5921 | **0.6861** | 0.01 |
| rel-amazon | user-churn | **0.7042** | 0.6643 | 0.6619 | 0.7039 | -0.04 |
| | item-churn | **0.8281** | 0.7797 | 0.7803 | 0.8255 | -0.31 |
| rel-stack | user-engagement | 0.9021 | 0.8847 | 0.8817 | **0.9053** | 0.35 |
| | user-badge | **0.8986** | 0.8608 | 0.8566 | 0.8632 | -3.94 |
| rel-hm | user-churn | **0.6988** | 0.6695 | 0.6569 | 0.6927 | -0.87 |

days or determining whether a clinical trial will achieve its primary outcome within the next year. In total, RelBench has 30 tasks across the 7 datasets, covering entity classification, entity regression, and recommendation. For our evaluation, we focus on 21 tasks on entity classification and regression [3].

## 4.1 SETUP AND BASELINES

We implement RELGT within the RDL pipeline (Robinson et al., 2024) by replacing the original GNN component, while preserving the learning mechanisms, database loaders, and task evaluators. The model has between 10-20M parameters, and we use a learning rate of $1e - 4$. For tasks with fewer than 1M training nodes, we tune the number of layers $L \in 1, 4, 8$ and use dropout rates of $0.3, 0.4, 0.5$. For tasks with more than 1M training nodes, we fix the number of layers to $L = 4$ due to compute budgets. For the sampling during the token preparation stage, we use $K = 300$ local neighbors and set $B = 4096$ as the number of tokens for global centroids. For smaller datasets (under one million training nodes), we use a batch size of 256 to ensure sufficient training steps. For larger datasets, we use a batch size of 1024. We do not perform exhaustive hyperparameter tuning; rather, our goal is to showcase the benefits of using RELGT in place of GNNs within the RDL framework. As shown in our ablation of the multi-element tokenization and global module in RELGT (Tab. 2), and context size (Fig. 4), careful tuning may further improve performance across different tasks.

In addition to the HeteroGNN baseline used in RDL, we report results for two variants of the Heterogeneous Graph Transformer (HGT) (Hu et al., 2020) to highlight the advantages of RELGT over existing GT models. Notably, many GTs, such as GraphGPS (Rampášek et al., 2022), are not directly applicable to heterogeneous graphs. Therefore, we adopt HGT and an enhanced version, HGT+PE, which incorporates Laplacian PE. These positional encodings are computed on the sampled subgraphs rather than the entire graph. Additional details are included in Appendix A.5.

## 4.2 RESULTS AND DISCUSSION

**RELGT improves over GNN in RDL.** The experimental results in Tables 1a and 1b demonstrate that RELGT consistently matches or outperforms the standard GNN baseline used in RDL (Robinson et al., 2024) across multiple datasets and tasks. Using a ±1% threshold to assess comparable performance, RELGT shows: (i) clear improvements (more than a 1% relative gain) on 10 tasks, (ii) comparable results (within ±1%) on 9 tasks, and (iii) competitive but lower performance (more than a 1% relative loss) on 2 tasks. We observe the largest improvements in `rel-trial site-success` (18.43%), `rel-avito ad-ctr` (15.85%), and `rel-f1 driver-top3` (10.56%), while on `rel-stack user-badge`, RELGT performs below the RDL baseline by a margin of -3.94%. For all other tasks, RELGT consistently improves or matches the performance of the baseline GNN. We attribute the overall performance improvement to two key factors: (i) the broader structural coverage enabled by RELGT's attention mechanisms as described in Section 3.2, and (ii) the fine-grained encodings employed in our tokenization scheme, which are further studied as follows and presented in Table 2.

---

[3] We exclude recommendation tasks in this work since they involve specific considerations, such as identifying target nodes (You et al., 2021) or using pair-wise learning architectures (Yuan et al., 2024). Details in Sec. A.1

Table 2: Relative drop (%) in performance in RELGT after removing a model component. Negative scores suggest the component is critical in RELGT, and vice-versa. Full results in Table 9.

| Dataset | Task | No Global Module | No GNN PE | No Node Type | No Hop Distance | No Relative Time |
|---------|------|------------------|-----------|--------------|-----------------|------------------|
| rel-avito | ad-ctr | −6.00 | −1.14 | −7.14 | −3.43 | −9.14 |
| rel-avito | user-clicks | 7.85 | −15.15 | 5.01 | 5.77 | 8.37 |
| rel-avito | user-visits | −0.35 | −2.38 | −0.11 | 0.39 | −0.75 |
| rel-event | user-ignore | −1.30 | 0.12 | −0.11 | 0.66 | −0.09 |
| rel-trial | study-outcome | −2.14 | −1.72 | 3.74 | −0.43 | 2.48 |
| rel-trial | site-success | −19.01 | −9.17 | −2.88 | −21.49 | −0.71 |
| rel-amazon | user-churn | −0.64 | −0.78 | 0.16 | 0.06 | −2.20 |
| rel-hm | item-sales | −9.33 | −17.35 | −12.69 | 0.93 | −77.24 |
| | **Average** | −3.87 | −5.95 | −1.75 | −2.19 | −9.91 |

**Subgraph GNN PE is critical in RELGT.** In Table 2, we highlight the importance of several components in RELGT by conducting ablation studies. We remove one component at a time while preserving all others, and report the relative performance drop compared to the full RELGT model. Our results show that removing the subgraph GNN (PE), which encodes local subgraph structure (Section 3.1), leads to consistent performance degradation across all tasks. This component proves critical for disambiguating parent-child relationships when full-attention is applied, thanks to the random node features initialization (Sato et al., 2021; Kanatsoulis et al., 2025). For instance, without the GNN (PE), *products* belonging to specific *transactions* (Figure 1) cannot be effectively captured, even when other encodings remain.

**Global module can bring gains depending on the task.** In the same Table 2, our results of removing the global attention to the learnable centroids (Section 3.2) reveal task-dependent patterns that align with the findings reported in (Kong et al., 2023; Dwivedi et al., 2023). For some tasks, such as rel-trial site-success, removing the attention to the centroids tokens leads to a substantial performance drop (-19.08%), indicating that the global database-wide context provides crucial information beyond the local neighborhood. However, for certain tasks such as rel-avito user-clicks, removing the global module actually improves performance (7.79% relative gain), suggesting that for some prediction targets, local information is sufficient, and the global context might introduce noise. These mixed results highlight the complementary nature of local and global information in relational graphs, with the latter being optional depending on the task.

**Ablation of other encodings.** The remaining ablations in Table 2 reveal mixed results across different components. While removing explicit fine-grained encodings (node type, hop distance, and relative time) degrades performance on some tasks, it improves performance on others. For tasks with specific temporal dependencies (detailed in Sec. A.1), our current temporal encodings may inadvertently introduce noise. Similarly, for node type and hop distance encodings, their information might already be partially captured by other model components. Despite these variations, the full RELGT model still shows consistently superior results when averaged across all tasks. However, our findings suggest that RELGT's scores could be further enhanced by careful tuning of these encoding components based on their task-specific importance. In particular, additional gains can be achieved by incorporating more effective temporal encodings (Clauset & Eagle, 2012; Huang et al., 2024; Jiang & Pu, 2023).

**HGT, a GT baseline, underperforms with significant overhead.** As shown in Tables 1a and 1b, HGT (Hu et al., 2020) underperforms compared to the HeteroGNN baseline of RDL (Robinson et al., 2024) across most tasks, with only two exceptions: rel-trial study-adverse and rel-event user-ignore. Notably, the integration of Laplacian PEs in HGT improves performance in 8 (of 21) tasks. Moreover, as illustrated in Figure 4, the computational overhead required for precomputing the Laplacian PEs substantially increases per-epoch runtime across various tasks. These empirical findings clearly reveal the difficulties of directly applying existing GT architectures to relational entity graphs, emphasizing the importance and need for our contributions with RELGT.

**Local context size $K$.** In our main RELGT experiments, we set the local context size at 300 nodes (Section 3.1), however, we study its variability in Figure 4 for context sizes $K \in \{100, 300, 500\}$. Although $K = 300$ generally produces the best results, optimal values vary across specific tasks. For instance, rel-avito ad-ctr benefits from a larger context size, whereas rel-trial study-outcome achieves better performance with a smaller context window. These findings

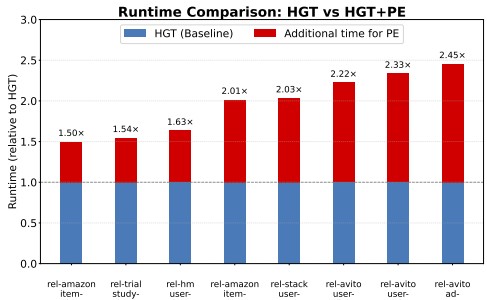 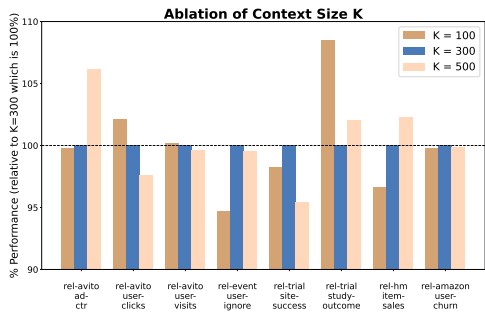

Figure 4: *Left:* Epoch runtime comparison of HGT (Hu et al., 2020) and HGT+PE, with Laplacian PE (see Figure 5 for all tasks). The red portion shows the additional time consumed by the precomputation of Laplacian PE against the base HGT time (blue). *Right:* Ablation for different $K$ values as the local context size in RELGT. Results using $K = 300$ serve as the baseline (100% performance), with $K = 100$ and $K = 500$ runs measured as % of performance relative to $K = 300$.

suggest that RELGT's performance could be further enhanced by task-specific tuning of the context size, allowing for better model expressivity based on the structural characteristics of each dataset.

## 5 CONCLUSION

In this work, we introduce the first Graph Transformer designed specifically for relational entity graphs: the Relational Graph Transformer. It addresses key challenges faced by existing models, such as incorporating heterogeneity, temporality, and comprehensive structural modeling within a unified GT framework. RELGT represents nodes as multi-element tokens enriched with fine-grained graph context and combines local attention over sampled subgraph tokens with global attention to learnable centroids, enabling effective representation learning on relational data. Experiments on the RelBench benchmark show that RELGT consistently outperforms GNN and GT baselines across multiple tasks. Moreover, our analysis highlights the critical role of subgraph-based positional encodings as a lightweight and effective alternative to traditional graph positional encodings. This work establishes RELGT as a powerful architecture for relational deep learning and opens new avenues for advancing and scaling such architectures toward foundation models tailored for relational data.

### ACKNOWLEDGMENTS

We thank Eric Chen, Shenyang Huang and Fang Wu for their helpful feedbacks and members of the Stanford SNAP group for their suggestions during the project. We also gratefully acknowledge the support of NSF under Nos. OAC-1835598 (CINES), CCF-1918940 (Expeditions), DMS-2327709 (IHBEM), IIS-2403318 (III); Stanford Data Applications Initiative, Wu Tsai Neurosciences Institute, Stanford Institute for Human-Centered AI, Chan Zuckerberg Initiative, Amazon, Genentech, Hitachi, and SAP. The content is solely the responsibility of the authors and does not necessarily represent the official views of the funding entities.

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

Table 3: Dataset and task statistics from RelBench used for our evaluation.

| Dataset | Task | Task type | #Rows of training table | | | #Unique Entities | %train/test Entity Overlap |
|---|---|---|---|---|---|---|---|
| | | | Train | Validation | Test | | |
| rel-amazon | user-churn | classification | 4,732,555 | 409,792 | 351,885 | 1,585,983 | 88.0 |
| | item-churn | classification | 2,559,264 | 177,689 | 166,842 | 416,352 | 93.1 |
| | user-ltv | regression | 4,732,555 | 409,792 | 351,885 | 1,585,983 | 88.0 |
| | item-ltv | regression | 2,707,679 | 166,978 | 178,334 | 427,537 | 93.5 |
| rel-avito | user-clicks | classification | 59,454 | 21,183 | 47,996 | 66,449 | 45.3 |
| | user-visits | classification | 86,619 | 29,979 | 36,129 | 63,405 | 64.6 |
| | ad-ctr | regression | 5,100 | 1,766 | 1,816 | 4,997 | 59.8 |
| rel-event | user-repeat | classification | 3,842 | 268 | 246 | 1,514 | 11.5 |
| | user-ignore | classification | 19,239 | 4,185 | 4,010 | 9,799 | 21.1 |
| | user-attendance | regression | 19,261 | 2,014 | 2,006 | 9,694 | 14.6 |
| rel-f1 | driver-dnf | classification | 11,411 | 566 | 702 | 821 | 50.0 |
| | driver-top3 | classification | 1,353 | 588 | 726 | 134 | 50.0 |
| | driver-position | regression | 7,453 | 499 | 760 | 826 | 44.6 |
| rel-hm | user-churn | classification | 3,871,410 | 76,556 | 74,575 | 1,002,984 | 89.7 |
| | item-sales | regression | 5,488,184 | 105,542 | 105,542 | 105,542 | 100.0 |
| rel-stack | user-engagement | classification | 1,360,850 | 85,838 | 88,137 | 88,137 | 97.4 |
| | user-badge | classification | 3,386,276 | 247,398 | 255,360 | 255,360 | 96.9 |
| | post-votes | regression | 2,453,921 | 156,216 | 160,903 | 160,903 | 97.1 |
| rel-trial | study-outcome | classification | 11,994 | 960 | 825 | 13,779 | 0.0 |
| | study-adverse | regression | 43,335 | 3,596 | 3,098 | 50,029 | 0.0 |
| | site-success | regression | 151,407 | 19,740 | 22,617 | 129,542 | 42.0 |

# A  APPENDIX

## A.1  BENCHMARK DETAILS

In this section, we include the details on the datasets and the tasks in RelBench (Robinson et al., 2024) which we use for our evaluation. RelBench consists of 7 datasets from diverse relational database domains, including e-commerce, clinical records, social networks, and sports, among others. These datasets are curated from their respective source domains and consist a wide range of sizes, from 1.3K to 5.4M records in the training set for the prediction tasks, with a total of 47M training records. For each dataset, multiple predictive tasks are defined, such as predicting a user's engagement with an advertisement within the next four days or determining whether a clinical trial will achieve its primary outcome within the next year. In total, RelBench has 30 tasks across the 7 datasets, covering entity classification, entity regression, and recommendation. For our evaluation, we focus on 21 tasks on entity classification and regression as RELGT primarily serves as a node representation learning model in RDL. We exclude recommendation tasks in this work since they involve specific considerations, such as identifying target nodes (You et al., 2021) or using pair-wise learning architectures (Yuan et al., 2024) and using RELGT trivially in RDL is sub-optimal. We detail the dataset and task statistics in Table 3.

### A.1.1  DATASETS

**rel-amazon.**  The Amazon E-commerce dataset consists of product details, user information, and review interactions from Amazon's platform, including metadata like pricing and categories, along with review ratings and content.

**rel-avito.**  Avito's marketplace dataset contains search queries, advertisement characteristics, and contextual information from this major online trading platform that facilitates transactions across various categories including real estate and vehicles.

**rel-event.**  The Event Recommendation dataset from Hangtime mobile app tracks users' social planning, capturing interactions, event details, demographic data, and social connections to reveal how relationships impact user behavior.

**rel-f1.**  The F1 dataset provides comprehensive Formula 1 racing information since 1950, documenting drivers, constructors, manufacturers, and circuits with detailed records of race results, standings, and specific data on various racing sessions and pit stops.

**rel-hm.** H&M's dataset contains customer-product interactions from their e-commerce platform, featuring customer demographics, product descriptions, and purchase histories.

**rel-stack.** The Stack Exchange dataset documents activity from this network of Q&A websites, including user biographies, posts, comments, edits, votes, and question relationships where users earn reputation through contributions.

**rel-trial.** The clinical trial dataset from the AACT initiative has study protocols and outcomes, containing trial designs, participant information, intervention details, and results metrics, serving as a key resource for medical research.

### A.1.2 TASKS

The following entity classification and regression tasks are defined in RelBench for the above datasets.

1. **rel-amazon**
   (a) `user-churn`: Predict whether a user will discontinue reviewing products within the next three months.
   (b) `item-churn`: Predict if a product will have no reviews in the next three months.
   (c) `user-ltv`: Estimate the total monetary value of merchandise in dolloar that a user will purchase and review within the next three months.
   (d) `item-ltv`: Estimate the total monetary value of purchases and reviews a product will receive during the next three months.

2. **rel-avito**
   (a) `user-visits`: Predict if a user will engage with several (advertisements) ads within the upcoming four days.
   (b) `user-clicks`: Predict whether a user will interact with multiple ads through clicking within the upcoming four days.
   (c) `ad-ctr`: Estimate the interaction probability for an ad, assuming it receives an interaction within four days.

3. **rel-event**
   (a) `user-attendance`: Estimate the number of of events a user will confirm attendance to (RSVP yes or maybe) within the upcoming seven days.
   (b) `user-repeat`: Predict whether a user will join an event (RSVP yes or maybe) within the upcoming seven days, provided they attended in an event during the previous fourteen days.
   (c) `user-ignore`: Predict whether a user will disregard or ignore more than two events invitations within the upcoming seven days.

4. **rel-f1**
   (a) `driver-dnf`: Predict if a driver will not finish a race within the upcoming month.
   (b) `driver-top3`: Determine if a driver will achieve a top-three qualifying position in a race within the upcoming month.
   (c) `driver-position`: Estimate a driver's average finishing placement across all races in the upcoming two months.

5. **rel-hm**
   (a) `user-churn`: Predict whether a customer will not perform any transactions in the upcoming week.
   (b) `item-sales`: Estimate total revenue generated by a product in the upcoming week.

6. **rel-stack**
   (a) `user-engagement`: Predict whether a user will contribute through voting, posting, or commenting within the upcoming three months.
   (b) `user-badge`: Predict whether a user will secure a new badge within the upcoming three months.

Table 4: Study of node initialization in Subgraph GNN PE. Relative drop is expressed as percentage drop of using $Z_{\text{LapPE}}$ vs. $Z_{\text{random}}$ and runtime ratio compares the time for $Z_{\text{LapPE}}$ vs. $Z_{\text{random}}$.

| Dataset | Task (# train) | MAE ↓ | Performance | | % Rel Drop | Epoch time (m) | | Runtime Ratio |
|---------|----------------|-------|-------------|------------|------------|----------------|------------|---------------|
| | | | $Z_{\text{random}}$ | $Z_{\text{LapPE}}$ | | $Z_{\text{random}}$ | $Z_{\text{LapPE}}$ | |
| rel-avito | ad-ctr | Test | **0.035** | 0.0369 | -5.43 | 0.76 | 2.57 | 3.38 |
| | | Val | 0.0314 | 0.0314 | | | | |
| rel-trial | site-success | Test | **0.326** | 0.3452 | -5.89 | 32.88 | 36.09 | 1.1 |
| | | Val | 0.359 | 0.3683 | | | | |
| rel-hm | item-sales | Test | **0.0536** | 0.0573 | -6.9 | 49.26 | 53.8 | 1.09 |
| | | Val | 0.0627 | 0.0667 | | | | |

| Dataset | Task (# train) | AUC ↑ | $Z_{\text{random}}$ | $Z_{\text{LapPE}}$ | % Rel Drop | $Z_{\text{random}}$ | $Z_{\text{LapPE}}$ | Runtime Ratio |
|---------|----------------|-------|-------------|------------|------------|----------------|------------|---------------|
| rel-avito | user-clicks | Test | **0.607** | 0.583 | -3.95 | 6.42 | 7.43 | 1.16 |
| | | Val | 0.656 | 0.6564 | | | | |
| | user-visits | Test | **0.664** | 0.6626 | -0.21 | 9.26 | 10.50 | 1.13 |
| | | Val | 0.699 | 0.7002 | | | | |
| rel-event | user-ignore | Test | **0.8** | 0.7988 | -0.15 | 1.85 | 2.77 | 1.5 |
| | | Val | 0.881 | 0.8916 | | | | |
| rel-trial | study-outcome | Test | **0.674** | 0.6532 | -3.09 | 1.41 | 1.52 | 1.08 |
| | | Val | 0.689 | 0.6719 | | | | |
| rel-amazon | user-churn | Test | 0.7039 | **0.7044** | 0.07 | 168.00 | 170.55 | 1.02 |
| | | Val | 0.7036 | 0.7036 | | | | |

   (c) `post-votes`: Estimate the number of votes a user's post will accumulate over the upcoming three months.

7. **rel-trial**

   (a) `study-outcome`: Predict whether a clinical trial will achieve its principal outcome within the upcoming year.

   (b) `study-adverse`: Estimate the number of patients who will experience significant adverse effects or mortality in a clinical trial over the upcoming year.

   (c) `site-success`: Estimate the success rate of a clinical trial site in the upcoming year.

## A.2 NODE INITIALIZATION FOR SUBGRAPH GNN PE IN RELGT

As described in Section 3.1, we employ a lightweight GNN PE to capture local graph structures that cannot be represented by other elements of the token, particularly the parent-child relationships among nodes in the local subgraph. The GNN is implemented as:

$$h_{\text{pe}}(v_j) = \text{GNN}(A_{\text{local}}, Z_{\text{random}})_j \in \mathbb{R}^d \tag{10}$$

where $\text{GNN}(\cdot, \cdot)_j$ is a lightweight GNN applied to the local subgraph, yielding the encoding for node $v_j$. Here, $A_{\text{local}} \in \mathbb{R}^{K \times K}$ represents the adjacency matrix of the sampled subgraph containing $K$ nodes, and $Z_{\text{random}} \in \mathbb{R}^{K \times d_{\text{init}}}$ denotes randomly initialized node features for the GNN (with $d_{\text{init}}$ as the initial feature dimension). In RELGT, we set $d_{\text{init}} = 1$.

The randomly initialized node features ($Z_{\text{random}}$) provide enhanced properties as discussed in Section 3.1. We investigate the alternative approach of using Laplacian PE ($Z_{\text{LapPE}}$) computed over the subgraph instead of random initialization and report these results in Table 4. For these results, we utilized a positional encoding dimension size of $4$. Our findings indicate that $Z_{\text{LapPE}}$ consistently underperforms compared to $Z_{\text{random}}$, while also introducing additional computational overhead ranging from $1.02\times$ to $3.38\times$ across the 8 selected tasks in our study. This shows the challenges of using existing PEs such as Laplacian PE in relational entity graphs and signify the use of GNN PE as part of RELGT's tokenization strategy.

## A.3 LEARNABLE SPATIO-TEMPORAL PE

In this section, we explore a learnable spatio-temporal positional encoding (PE) for RELGT. Instead of using the relative time encoder (Eqn. 4), we use the 'relative time' term to initialize nodes in

Table 5: Performance comparison of RELGT (Full) with the Spatio-Temporal PE (Eqns. 11 - 12). Negative scores suggest performance drop with the spatio-temporal PE in RELGT.

| Dataset | Task | MAE ↓ | | RelGT (Full) | RelGT (Spatio-Temporal PE) | % Rel. Diff |
|---|---|---|---|---|---|---|
| rel-avito | ad-ctr | Test | | **0.0345** | 0.0355 | -2.90 |
| | | Val | | 0.0314 | 0.0315 | |
| rel-trial | site-success | Test | | **0.3262** | 0.3554 | -8.95 |
| | | Val | | 0.3593 | 0.3883 | |
| rel-hm | item-sales | Test | | **0.0536** | 0.0630 | -17.54 |
| | | Val | | 0.0627 | 0.0718 | |

| Dataset | Task | AUC ↑ | | RelGT (Full) | RelGT (Spatio-Temporal PE) | % Rel. Diff |
|---|---|---|---|---|---|---|
| rel-avito | user-clicks | Test | | **0.6830** | 0.6465 | -5.34 |
| | | Val | | 0.6649 | 0.6519 | |
| | user-visits | Test | | **0.6678** | 0.6641 | -0.55 |
| | | Val | | 0.7024 | 0.7017 | |
| rel-event | user-ignore | Test | | **0.8157** | 0.8152 | -0.06 |
| | | Val | | 0.8868 | 0.8870 | |
| rel-trial | study-outcome | Test | | **0.6861** | 0.6537 | -4.72 |
| | | Val | | 0.6678 | 0.6757 | |
| rel-amazon | user-churn | Test | | **0.7039** | 0.7036 | -0.04 |
| | | Val | | 0.7036 | 0.7037 | |

Table 6: RELGT results on entity classification tasks in RelBench compared with Griffin (Wang et al., 2025). AUC is the performance metric. Higher is better.

| Dataset | Task | Griffin | RelGT (ours) | % Rel. Gain |
|---|---|---|---|---|
| rel-f1 | driver-dnf | 0.745 | **0.7587** | 1.84 |
| | driver-top3 | 0.825 | **0.8352** | 1.24 |
| rel-avito | user-clicks | 0.630 | **0.6830** | 8.41 |
| | user-visits | 0.650 | **0.6678** | 2.74 |
| rel-trial | study-outcome | **0.689** | 0.6861 | -0.42 |
| rel-amazon | user-churn | 0.700 | **0.7039** | 0.56 |
| | item-churn | 0.811 | **0.8255** | 1.79 |
| rel-stack | user-engagement | 0.898 | **0.9053** | 0.81 |
| | user-badge | **0.870** | 0.8632 | -0.78 |
| rel-hm | user-churn | 0.683 | **0.6927** | 1.42 |

the Subgraph GNN PE, where relative time $\tau(v_j) - \tau(v_i)$ denotes the temporal difference between neighbor node $v_j$ and seed node $v_i$. This approach repurposes the Subgraph GNN PE as a learnable spatio-temporal PE, which is defined as:

$$h_{\text{stpe}}(v_j) = \text{GNN}(A_{\text{local}}, Z_{\text{relative\_time}})_j \in \mathbb{R}^d \tag{11}$$

where $z_{j,\text{relative\_time}} = \tau(v_j) - \tau(v_i)$. The token representation, then, becomes:

$$h_{\text{token}}(v_j) = O \cdot [h_{\text{feat}}(v_j) \,||\, h_{\text{type}}(v_j) \,||\, h_{\text{hop}}(v_i, v_j) \,||\, h_{\text{stpe}}(v_j)] \tag{12}$$

where $||$ denotes the concatenation of the individual encoder outputs, and $O \in \mathbb{R}^{4d \times d}$ is a learnable matrix to mix the embeddings.

Table 5 presents our evaluation results over three regression and five classification tasks in RelBench. Across all tasks, replacing the original temporal encoder and subgraph GNN PE encoder with the Spatio-Temporal PE leads to a consistent performance decline in RELGT.

## A.4 COMPARISON WITH GRIFFIN

In this section, we compare RELGT with Griffin (Wang et al., 2025), which is a GNN based relational foundation model that integrates unified feature encoders, cross-attention and hierarchical message passing to process relational entity graphs from diverse domains. We report the results in Table 6

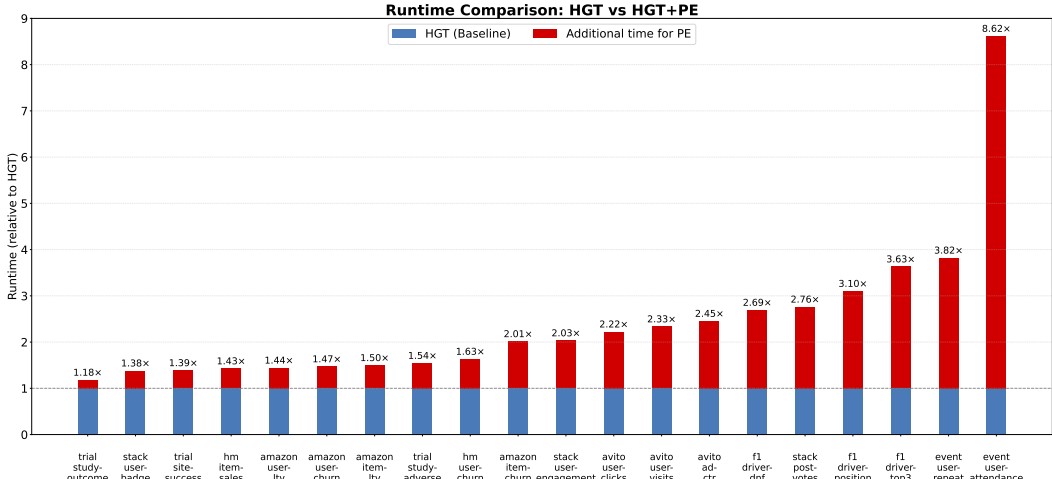

Figure 5: Runtime Comparison of HGT and HGT+PE baseline. Adding the Laplacian Positional Encoding increases computational overhead, with penalties on average training time per epoch. The overhead for PE reaches up to 761% relative to the training time of HGT on the same dataset.

for 10 entity classification tasks in RelBench, where Griffin is finetuned on each task following the same procedure used to train RELGT. RelGT outperforms Griffin on 8 out of 10 tasks with relative gains of up to 8.41%, demonstrating the advantages of a Graph Transformer backbone for processing relational entity graphs.

## A.5 HGT BASELINE

In the main experiments (Section 4), we use the Heterogeneous Graph Transformer (HGT) (Hu et al., 2020) as a graph transformer (GT) baseline, and report results for two variants to demonstrate the advantages of RELGT over existing GT models. Specifically, we consider the standard HGT model and an enhanced version, HGT+PE, which incorporates Laplacian positional encodings (LapPE). These positional encodings are computed on sampled subgraphs rather than the full graph.

For implementation, we use the `HGTConv` layer from PyTorch Geometric (Fey & Lenssen, 2019) and integrate it into the RDL pipeline (Robinson et al., 2024) by replacing the default GNN module. Both variants use 4 attention heads and 2 layers, similar to the configuration of the GNN module in RDL, with residual connections and layer normalization applied between layers. For the HGT+PE variant, we use LapPE of dimension 4 for all tasks, except for `rel-amazon item-ltv` and `rel-hm item-sales`, where we use dimension 2. Notably, because the relational entity graphs are heterogeneous, the Laplacian positional encodings is computed multiple times for each node type, unlike the original homogeneous setting for which LapPE was designed (Dwivedi et al., 2020).

In addition to the main results in Table 1, we report per-epoch runtimes in Figure 5 and Table 7. We observe a significant computational overhead from precomputing Laplacian positional encodings, with slowdowns ranging from 1.8× to 8.62×, highlighting the challenge of directly applying existing graph PE techniques *as is* to relational entity graphs, and signifying the contributions of RELGT.

## A.6 DETAILED RESULTS

In Table 8, we report the full results of different configurations we tuned for RELGT, particularly on the smaller datasets with lesser than a million training nodes. Table 9 provides the full scores for the RELGT component study in Table 2, while Table 10 provides the supporting results for Figure 4. Finally, we provide the elaborated version of the Tables 1a and 1b in Tables 11 and 12, respectively.

Table 7: Relative performance drop (%) when position encoding (PE) is removed from HGT+PE models and average training time per epoch of HGT and HGT+PE. Negative scores suggest the PE is critical, and vice-versa. HGT+PE consistently requires more training time per epoch compared to HGT without PE across all datasets.

| Dataset | Task | No PE | HGT(s) | HGT+PE(s) |
|---------|------|-------|--------|-----------|
| rel-f1 | driver-position | 1.79 | 1.47 | 4.56 |
| rel-avito | ad-ctr | 10.73 | 1.63 | 4.00 |
| rel-event | user-attendance | −2.85 | 4.36 | 37.57 |
| rel-trial | study-adverse | −2.03 | 9.72 | 15.02 |
| rel-trial | site-success | 1.29 | 45.73 | 63.41 |
| rel-amazon | user-ltv | 3.45 | 73.59 | 106.21 |
| rel-amazon | item-ltv | −0.93 | 73.68 | 110.33 |
| rel-stack | post-votes | 0.15 | 191.23 | 528.25 |
| rel-hm | item-sales | −2.18 | 94.66 | 135.05 |
| rel-f1 | driver-dnf | 0.46 | 2.54 | 6.84 |
| rel-f1 | driver-top3 | −23.39 | 0.38 | 1.38 |
| rel-avito | user-clicks | 3.08 | 11.09 | 24.66 |
| rel-avito | user-visits | −1.24 | 17.16 | 40.07 |
| rel-event | user-repeat | 1.93 | 1.35 | 5.16 |
| rel-event | user-ignore | 2.29 | 4.49 | 651.10 |
| rel-trial | study-outcome | −0.21 | 4.09 | 4.83 |
| rel-amazon | user-churn | 0.29 | 78.56 | 115.53 |
| rel-amazon | item-churn | −0.20 | 75.51 | 152.06 |
| rel-stack | user-engagement | 0.52 | 175.16 | 356.07 |
| rel-stack | user-badge | 1.57 | 153.68 | 212.21 |
| rel-hm | user-churn | 4.34 | 77.73 | 127.04 |
| | **Average** | −0.05 | 52.28 | 128.64 |

## A.7 COMPUTATIONAL COMPLEXITY AND RESOURCE INFORMATION.

RELGT has $O(K^2 \cdot d)$ complexity for local attention and $O(K \cdot B \cdot d)$ for global attention per node, where $K$ is local context size, $B$ is number of global centroids, and $d$ is hidden dimension. We implement RELGT using PyTorch framework (Paszke, 2019), PyTorch Geometric framework (Fey & Lenssen, 2019) and adapt the codebase of relational deep learning (Robinson et al., 2024) https://github.com/snap-stanford/relbench. All our experiments are conducted on an NVIDIA A100 GPU server with 8 GPU nodes.

Table 8: RELGT results using $L \in 1, 4, 8$ and dropout $\in 0.3, 0.4, 0.5$ for the smaller datasets with less than a million training nodes.

| Dataset | Task (# train) | MAE ↓ | L1 0.3 | L1 0.4 | L1 0.5 | L4 0.3 | L4 0.4 | L4 0.5 | L8 0.3 | L8 0.4 | L8 0.5 |
|---------|----------------|-------|--------|--------|--------|--------|--------|--------|--------|--------|--------|
| rel-f1 | driver-position (7k) | Test | 4.942 | 5.6431 | **3.917** | 4.6316 | 4.0851 | 4.0042 | 5.5273 | 5.5569 | 4.6085 |
| | | Val | 3.1897 | 3.1817 | 3.3257 | 3.1046 | 3.3352 | 3.1276 | 3.1589 | 3.2907 | 3.1843 |
| rel-avito | ad-ctr (5k) | Test | 0.0358 | 0.0352 | **0.0345** | 0.035 | 0.0366 | 0.038 | 0.0354 | 0.0358 | 0.0356 |
| | | Val | 0.0322 | 0.0313 | 0.0314 | 0.0322 | 0.0335 | 0.0317 | 0.0322 | 0.0324 | |
| rel-event | user-attendance (19k) | Test | 0.2635 | 0.2595 | 0.2635 | **0.2502** | 0.2543 | 0.2584 | 0.2635 | 0.2637 | 0.2635 |
| | | Val | 0.2618 | 0.2558 | 0.2618 | 0.2548 | 0.2534 | 0.253 | 0.2618 | 0.2599 | 0.2618 |
| rel-trial | study-adverse (43k) | Test | 44.8553 | 44.2260 | 44.848 | 44.8893 | 44.4310 | **43.9923** | 44.2245 | 44.5878 | 44.5013 |
| | | Val | 46.3538 | 46.3193 | 46.2056 | 46.1031 | 45.9498 | 46.2148 | 46.1804 | 46.1381 | 46.4332 |
| | site-success (151k) | Test | 0.3490 | 0.3652 | 0.3830 | 0.4019 | 0.386 | **0.3262** | 0.3783 | 0.3431 | 0.3644 |
| | | Val | 0.3493 | 0.3455 | 0.3550 | 0.3771 | 0.392 | 0.3593 | 0.3848 | 0.3643 | 0.3669 |

| Dataset | Task (# train) | AUC ↑ | L1 0.3 | L1 0.4 | L1 0.5 | L4 0.3 | L4 0.4 | L4 0.5 | L8 0.3 | L8 0.4 | L8 0.5 |
|---------|----------------|-------|--------|--------|--------|--------|--------|--------|--------|--------|--------|
| rel-f1 | driver-dnf (11k) | Test | 0.7434 | 0.7587 | 0.7521 | **0.7587** | 0.745 | 0.6957 | 0.7349 | 0.7393 | 0.741 |
| | | Val | 0.6877 | 0.6761 | 0.6896 | 0.6804 | 0.6762 | 0.6768 | 0.6702 | 0.6803 | 0.6865 |
| | driver-top3 (1k) | Test | 0.7845 | 0.8203 | 0.8 | 0.8171 | 0.8157 | **0.8352** | 0.7871 | 0.8217 | 0.8222 |
| | | Val | 0.7775 | 0.783 | 0.7764 | 0.7841 | 0.79 | 0.7958 | 0.7893 | 0.7847 | 0.7829 |
| rel-avito | user-clicks (59k) | Test | 0.6524 | 0.6233 | 0.6212 | 0.6067 | 0.5893 | 0.596 | 0.6245 | **0.683** | 0.6507 |
| | | Val | 0.6649 | 0.6616 | 0.6501 | 0.6564 | 0.6608 | 0.6579 | 0.6587 | 0.6649 | 0.6648 |
| | user-visits (86k) | Test | 0.6627 | 0.6663 | 0.665 | 0.6615 | 0.6584 | 0.6642 | 0.6647 | **0.6678** | 0.664 |
| | | Val | 0.7005 | 0.6993 | 0.7001 | 0.6954 | 0.6958 | 0.699 | 0.6995 | 0.7024 | 0.7011 |
| rel-event | user-repeat (3k) | Test | 0.6981 | 0.7403 | 0.7452 | 0.7563 | 0.7236 | 0.7432 | **0.7609** | 0.7316 | 0.7418 |
| | | Val | 0.7172 | 0.7386 | 0.7319 | 0.7245 | 0.7207 | 0.736 | 0.7285 | 0.7209 | 0.7064 |
| | user-ignore (19k) | Test | 0.8006 | 0.802 | 0.7986 | 0.799 | 0.787 | 0.8002 | 0.7956 | 0.8076 | **0.8157** |
| | | Val | 0.8739 | 0.8721 | 0.8729 | 0.878 | 0.8731 | 0.881 | 0.8757 | 0.8801 | 0.8868 |
| rel-trial | study-outcome (11k) | Test | 0.6808 | 0.6753 | 0.6837 | 0.6488 | 0.6818 | 0.6744 | **0.6861** | 0.6562 | 0.6649 |
| | | Val | 0.6815 | 0.6792 | 0.6751 | 0.6737 | 0.676 | 0.689 | 0.6678 | 0.6746 | 0.6768 |

Table 9: Relative drop (%) in performance in RELGT after removing a model component. Negative scores suggest the component is critical in RELGT, and vice-versa.

| Dataset | Task (# train) | MAE↓ | RelGT (Full) | RelGT (No Global) | % Rel. Drop | RelGT (No GNN) | % Rel. Drop | RelGT (No Type) | % Rel. Drop | RelGT (No Hop) | % Rel. Drop | RelGT (No Time) | % Rel. Drop |
|---|---|---|---|---|---|---|---|---|---|---|---|---|---|
| rel-avito | ad-ctr | Test | **0.0350** | 0.0371 | -6.0 | 0.0354 | -1.14 | 0.0375 | -7.14 | 0.0362 | -3.43 | 0.0382 | -9.14 |
| | | Val | 0.0314 | 0.0323 | | 0.0315 | | 0.0328 | | 0.0322 | | 0.0337 | |
| rel-trial | site-success | Test | **0.3262** | 0.3882 | -19.01 | 0.3561 | -9.17 | 0.3356 | -2.88 | 0.3963 | -21.49 | 0.3285 | -0.71 |
| | | Val | 0.3593 | 0.3342 | | 0.3637 | | 0.3655 | | 0.3614 | | 0.3615 | |
| rel-hm | item-sales | Test | 0.0536 | 0.0586 | -9.33 | 0.0629 | -17.35 | 0.0604 | -12.69 | **0.0531** | 0.93 | 0.095 | -77.24 |
| | | Val | 0.0627 | 0.0676 | | 0.073 | | 0.0696 | | 0.0623 | | 0.1025 | |

| Dataset | Task (# train) | AUC↑ | RelGT (Full) | RelGT (No Global) | % Rel. Drop | RelGT (No GNN) | % Rel. Drop | RelGT (No Type) | % Rel. Drop | RelGT (No Hop) | % Rel. Drop | RelGT (No Time) | % Rel. Drop |
|---|---|---|---|---|---|---|---|---|---|---|---|---|---|
| rel-avito | user-clicks | Test | 0.6067 | 0.6543 | 7.85 | 0.5148 | -15.15 | 0.6371 | 5.01 | 0.6417 | 5.77 | **0.6575** | 8.37 |
| | | Val | 0.6564 | 0.6496 | | 0.6551 | | 0.6559 | | 0.6482 | | 0.6579 | |
| | user-visits | Test | 0.6642 | 0.6619 | -0.35 | 0.6484 | -2.38 | 0.6635 | -0.11 | **0.6668** | 0.39 | 0.6592 | -0.75 |
| | | Val | 0.699 | 0.6892 | | 0.6879 | | 0.6991 | | 0.7016 | | 0.7005 | |
| rel-event | user-ignore | Test | 0.8002 | 0.7898 | -1.3 | 0.8012 | 0.12 | 0.7993 | -0.11 | **0.8055** | 0.66 | 0.7995 | -0.09 |
| | | Val | 0.881 | 0.8575 | | 0.8637 | | 0.8873 | | 0.8852 | | 0.8789 | |
| rel-trial | study-outcome | Test | 0.6744 | 0.66 | -2.14 | 0.6628 | -1.72 | **0.6996** | 3.74 | 0.6715 | -0.43 | 0.6911 | 2.48 |
| | | Val | 0.689 | 0.664 | | 0.6775 | | 0.6728 | | 0.6705 | | 0.6578 | |
| rel-amazon | user-churn | Test | 0.7039 | 0.6994 | -0.64 | 0.6984 | -0.78 | **0.705** | 0.16 | 0.7043 | 0.06 | 0.6884 | -2.2 |
| | | Val | 0.7036 | 0.6994 | | 0.6994 | | 0.7042 | | 0.704 | | 0.6882 | |

Table 10: Ablation of context size $K$ in RELGT.

| Dataset | Task (# train) | MAE↓ | RELGT K=100 | RELGT K=300 | RELGT K=500 |
|---|---|---|---|---|---|
| rel-avito | ad-ctr | Test | 0.0375 | 0.0374 | **0.0351** |
| | | Val | 0.0329 | 0.0319 | 0.031 |
| rel-trial | site-success | Test | 0.3739 | **0.3674** | 0.3842 |
| | | Val | 0.3708 | 0.372 | 0.376 |
| rel-hm | item-sales | Test | 0.055 | 0.0532 | **0.052** |
| | | Val | 0.0643 | 0.0619 | 0.061 |

| Dataset | Task (# train) | AUC↑ | RELGT K=100 | RELGT K=300 | RELGT K=500 |
|---|---|---|---|---|---|
| rel-avito | user-clicks | Test | **0.6628** | 0.6491 | 0.6334 |
| | | Val | 0.6437 | 0.6622 | 0.6632 |
| | user-visits | Test | **0.6664** | 0.6653 | 0.6627 |
| | | Val | 0.7013 | 0.701 | 0.7005 |
| rel-event | user-ignore | Test | 0.7674 | **0.8105** | 0.8068 |
| | | Val | 0.8682 | 0.8853 | 0.8843 |
| rel-trial | study-outcome | Test | **0.7078** | 0.6526 | 0.666 |
| | | Val | 0.6575 | 0.663 | 0.6877 |
| rel-amazon | user-churn | Test | 0.7038 | **0.7054** | 0.7043 |
| | | Val | 0.7033 | 0.7044 | 0.7042 |

Table 11: Results on the entity regression tasks in RelBench. Lower is better. Best values are in **bold**. Relative gains are expressed as percentage improvement over RDL baseline.

| Dataset | Task | MAE↓ | RDL Baseline | HGT | HGT+PE | RelGT (ours) | % Rel. Gain |
|---|---|---|---|---|---|---|---|
| rel-f1 | driver-position | Test | 4.022±0.119 | 4.2263±0.0580 | 4.3921±0.1382 | **3.9170**±0.3448 | 2.61 |
| | | Val | 3.193±0.024 | 3.1543±0.1455 | 3.1116±0.1120 | 3.3257±0.5618 | |
| rel-avito | ad-ctr | Test | 0.041±0.001 | 0.0462±0.0021 | 0.0483±0.0027 | **0.0345**±0.0009 | 15.85 |
| | | Val | 0.037±0.000 | 0.0433±0.0019 | 0.0444±0.0024 | 0.0314±0.0010 | |
| rel-event | user-attendance | Test | 0.258±0.006 | 0.2635±0.0000 | 0.2611±0.0043 | **0.2502**±0.0033 | 2.79 |
| | | Val | 0.255±0.007 | 0.2616±0.0001 | 0.2603±0.0020 | 0.2548±0.0018 | |
| rel-trial | study-adverse | Test | 44.473±0.209 | 45.1692±2.6927 | **42.6484**±0.2785 | 43.9923±0.5928 | 1.08 |
| | | Val | 46.290±0.304 | 47.3913±1.7936 | 45.7910±0.0051 | 46.2148±0.7210 | |
| | site-success | Test | 0.400±0.020 | 0.4428±0.0047 | 0.4396±0.0083 | **0.3263**±0.0306 | 18.43 |
| | | Val | 0.401±0.009 | 0.4275±0.0062 | 0.4292±0.0069 | 0.3593±0.0372 | |
| rel-amazon | user-ltv | Test | 14.313±0.013 | 15.4120±0.0447 | 15.8643±0.0924 | **14.2665**±0.0154 | 0.32 |
| | | Val | 12.132±0.007 | 13.2295±0.1402 | 13.4886±0.0713 | 12.1151±0.0218 | |
| | item-ltv | Test | 50.053±0.163 | 55.8683±0.6003 | 55.8493±0.3226 | **48.9222**±0.7006 | 2.26 |
| | | Val | 45.140±0.068 | 51.0303±0.2230 | 50.6522±0.6141 | 43.8161±0.0548 | |
| rel-stack | post-votes | Test | **0.065**±0.000 | 0.0679±0.0000 | 0.0680±0.0000 | 0.0654±0.0002 | -0.62 |
| | | Val | 0.059±0.000 | 0.0615±0.0000 | 0.0617±0.0000 | 0.0592±0.0002 | |
| rel-hm | item-sales | Test | 0.056±0.000 | 0.0641±0.0012 | 0.0639±0.0003 | **0.0536**±0.0006 | 4.29 |
| | | Val | 0.065±0.000 | 0.0739±0.0008 | 0.0739±0.0010 | 0.0627±0.0008 | |

Table 12: Results on the entity classification tasks in RelBench. Higher is better. Best values are in **bold**. Relative gains are expressed as percentage improvement over RDL baseline.

| Dataset | Task | AUC ↑ | RDL Baseline | HGT | HGT +PE | RelGT (ours) | % Rel. Gain |
|---------|------|-------|--------------|-----|---------|--------------|-------------|
| rel-f1 | driver-dnf | Test | 0.7262±0.0027 | 0.7077±0.0153 | 0.7117±0.0084 | **0.7587**±0.0413 | 4.48 |
| | | Val | 0.7136±0.0154 | 0.7765±0.0066 | 0.7340±0.0018 | 0.6804±0.0420 | |
| | driver-top3 | Test | 0.7554±0.0063 | 0.7075±0.1156 | 0.7627±0.1390 | **0.8352**±0.0342 | 10.56 |
| | | Val | 0.7764±0.0316 | 0.6457±0.0147 | 0.6486±0.0287 | 0.7958±0.0513 | |
| rel-avito | user-clicks | Test | 0.6590±0.0195 | 0.6376±0.0298 | 0.6457±0.0099 | **0.6830**±0.0602 | 3.64 |
| | | Val | 0.6473±0.0032 | 0.5999±0.0022 | 0.5886±0.0231 | 0.6649±0.0610 | |
| | user-visits | Test | 0.6620±0.0010 | 0.6432±0.0002 | 0.6495±0.0022 | **0.6678**±0.0015 | 0.88 |
| | | Val | 0.6965±0.0004 | 0.6652±0.0040 | 0.6649±0.0060 | 0.7024±0.0009 | |
| rel-event | user-repeat | Test | **0.7689**±0.0159 | 0.6496±0.0220 | 0.6536±0.0137 | 0.7609±0.0219 | -1.04 |
| | | Val | 0.7125±0.0253 | 0.6082±0.0148 | 0.6148±0.0172 | 0.7285±0.0108 | |
| | user-ignore | Test | 0.8162±0.0111 | **0.8247**±0.0096 | 0.8161±0.0001 | 0.8157±0.0040 | -0.06 |
| | | Val | 0.9170±0.0033 | 0.8997±0.0114 | 0.8940±0.0000 | 0.8868±0.0032 | |
| rel-trial | study-outcome | Test | 0.6860±0.0101 | 0.5837±0.0141 | 0.5921±0.0303 | **0.6861**±0.0040 | 0.01 |
| | | Val | 0.6818±0.0049 | 0.6037±0.0040 | 0.6025±0.0071 | 0.6678±0.0038 | |
| rel-amazon | user-churn | Test | **0.7042**±0.0005 | 0.6643±0.0041 | 0.6619±0.0042 | 0.7039±0.0008 | -0.04 |
| | | Val | 0.7045±0.0006 | 0.6680±0.0029 | 0.6652±0.0030 | 0.7036±0.0008 | |
| | item-churn | Test | **0.8281**±0.0003 | 0.7797±0.0039 | 0.7803±0.0053 | 0.8255±0.0006 | -0.31 |
| | | Val | 0.8239±0.0002 | 0.7816±0.0031 | 0.7803±0.0030 | 0.8220±0.0010 | |
| rel-stack | user-engagement | Test | 0.9021±0.0007 | 0.8847±0.0044 | 0.8817±0.0046 | **0.9053**±0.0005 | 0.35 |
| | | Val | 0.9059±0.0009 | 0.8863±0.0039 | 0.8811±0.0034 | 0.9033±0.0013 | |
| | user-badge | Test | **0.8986**±0.0008 | 0.8608±0.0044 | 0.8566±0.0068 | 0.8632±0.0018 | -3.94 |
| | | Val | 0.8886±0.0008 | 0.8732±0.0025 | 0.8710±0.0016 | 0.8741±0.0050 | |
| rel-hm | user-churn | Test | **0.6988**±0.0021 | 0.6695±0.0067 | 0.6569±0.0109 | 0.6927±0.0019 | -0.87 |
| | | Val | 0.7042±0.0009 | 0.6727±0.0062 | 0.6605±0.0103 | 0.6988±0.0034 | |

