# OpenReview forum: "Relational Graph Transformer"
_ICLR.cc/2026/Conference — ICLR 2026 Poster_

### Official Review · Reviewer_nFvD · 2025-10-16

**Soundness:** 2
**Presentation:** 3
**Contribution:** 1
**Rating:** 2
**Confidence:** 5

**Summary:**

This paper introduces RELGT (Relational Graph Transformer), a novel Transformer architecture designed specifically for Relational Deep Learning (RDL),  i.e., learning directly from multi-table relational databases represented as heterogeneous temporal graphs. The model tokenizes each node into five elements: features, type, hop distance, time difference, and subgraph-based positional encoding, aiming to capture heterogeneity, temporality, and structural complexity. RELGT combines local attention (over sampled subgraphs) and global attention (to learnable centroids), and is evaluated on 21 tasks from the RelBench benchmark, outperforming heterogeneous GNNs (HeteroGNN) and HGT baselines by up to 18%.

**Strengths:**

- Ambitious motivation: Addresses an important gap, the adaptation of Transformers to relational databases, an increasingly relevant setting (RDL).
- Unified treatment of temporal, heterogeneous, and structural properties within a single architecture.
- Comprehensive benchmark evaluation on RelBench (21 tasks), with reasonable performance improvements and ablations to justify design choices.
- The multi-element tokenization idea (features, type, hop, time, subgraph PE) provides a structured approach to encode complex relational contexts.
- The local–global hybrid design aligns with recent scalable GT approaches and avoids expensive positional encoding precomputation.
Ablation studies are thorough and help identify which token elements contribute most.

**Weaknesses:**

- Novelty is overstated.
   - Similar approaches already exist, e.g. Heterogeneous Temporal Graph Transformers and other models combining subgraph tokenization with structural encoders. The paper presents RELGT as “the first GT for relational graphs,” but other prior works [1-3] already tackle heterogeneity with a transformer .
   - The tokenization strategy appears as a straightforward concatenation of known encodings (type, hop, time, PE), rather than a fundamentally new design.
  - The subgraph GNN PE encoder appears conceptually very similar to the approach proposed in [4], as also acknowledged by the authors.

- Conceptual gaps / unclear definitions.
   - The notion of “schema-defined structure” in 2.1 is not clearly formalized. Other heterogeneous graphs (DBLP, IMDB, ACM) also have schema-defined relations, so this does not uniquely characterize relational data.

- Overstatements and misleading claims.
   - The paper states that RDL GNNs outperform LightGBM on all tasks, while the original RDL paper shows a few opposite results.
   - Although the abstract claims that RELGT “consistently matches or outperforms” GNN baselines, this statement is not supported by the reported results. In Table 1b (entity classification), the proposed model performs worse than the RDL baseline in about half of the tasks, with gains often close to zero and only a few cases showing meaningful improvements. In Table 1a (regression), the trend is more favorable to the proposed model, although it would be more informative to report the gains in absolute rather than percentage terms.
   - In Section 4.2, the authors state that HGT underperforms even compared to RDL, and they provide a time complexity analysis showing that adding Laplacian Encodings (LE) to HGT improves performance but at the cost of higher runtime. However, what is missing is a temporal (runtime) analysis of the proposed model compared to all competitors. Moreover, the HGT method has been improved in paper [3], specifically addressing the runtime efficiency issue, but this variant was not considered, making the temporal analysis incomplete.
   - Furthermore, the reported results (of model and competitors) do not include mean and standard deviation across multiple runs, which are crucial to assess statistical significance and reproducibility of the claimed improvements, as explained in [4].

- Technical novelty and clarity.
   - Most encoders (feature, type, hop, time, PE) are standard components and not surprising.
   - The model essentially applies a Transformer on subgraph samples with concatenated encodings, which is conceptually simple and not theoretically grounded.
   - Lack of formal analysis of computational complexity, scalability, or positional encoding stability.

- Missing broader context.
   - No discussion on relational inductive bias or how RELGT generalizes to unseen tables or relations.
   - No exploration of few-shot or inductive RDL settings where Transformers could excel.

[1] Wang et al. HTGformer: Heterogeneous Temporal Graph Transformer

[2] Hu et al, Heterogeneous Graph Transformer

[3] Yun et al, Graph transformer networks: Learning meta-path graphs to improve gnns.

[4] Demšar et al., Statistical Comparisons of Classifiers over Multiple Data Sets

**Questions:**

- How does RELGT differ concretely and conceptually from existing Heterogeneous Temporal Graph Transformers?
- Can the authors clarify what they mean by “schema-defined structure” and how it is different from the one of other heterogeneous graph datasets like IMDB, DBLP, ACM and why this structure requires a special architecture?
- How is the subgraph GNN PE trained and does it introduce additional computational cost?
- Why other multi-relational GNN models [5-8] have not been tested even if they should be able to work in this multi-relational setting?
- Could the authors quantify the scalability and training time of RELGT compared to RDL and other competitors?


[5] Schlichtkrull et al., Modeling relational data with graph convolutional networks

[6] Yuet al.,  Heterogeneous graph representation learning with relation awareness.

[7] Zhu et al.,  Relation structure-aware heterogeneous graph neural network.

[8] Ferrini et al., A Self-Explainable Heterogeneous GNN for Relational Deep Learning

---

> ### Author Response · Authors · 2025-11-22
>
> We thank the reviewer for taking the time reading and evaluating our work, and for the suggestions on improving our manuscript. We address below the key concerns raised in the review.
>
> **1. On Novelty: works for relational graphs, tokenization, and subgraph GNN PE**
> **Authors**: We respectfully disagree with the characterization that our novelty is overstated, and we address each point below:
> **(i) "First GT for relational graphs"**: We acknowledge, review and compare that prior works tackle heterogeneity with transformers. However, none of these simultaneously address the unique combination of challenges in relational entity graphs: heterogeneity, temporality, AND schema-defined structures from relational databases. Specifically:
> - HGT handles heterogeneity but lacks temporal modeling and graph PEs. Other heterogeneous transformers similarly don't address the temporal dimension or the specific constraints of database schemas
> - HTGT handles temporality and heterogeneity separately with two different modules and works iteratively. In addition, it does not leverage graph PE components which have been critical in the development of GTs.
> - No prior work has demonstrated effective transformer architectures specifically for the RelBench RDL setting
>
> In addition, the terms “relational graphs” and “relational entity graphs” in the context of this work refer to the graphs derived from relational databases, which is defined formally in Section 2.1.
>
> **(ii) "Straightforward concatenation"**: We argue this critique misses the key insight of our work and refer to Section 3.1 where we discuss the background for this. In particular, standard Transformers represent tokens using two components: token features and positional structure. Graph Transformers follow the same abstraction, relying on node features and graph positional encodings (PEs). However, existing PEs are not designed to capture the unique complexities of relational entity graphs, i.e., heterogeneity, temporality, and schema-defined structures, which motivates our design. Our multi-element tokenization integrates multiple complementary aspects of relational structure that are difficult to encode within the traditional two-element framework. The ablations in Table 2 highlight that each element addresses specific modeling limitations, with certain components (e.g., GNN PE) being especially critical, while others exhibit task-dependent impact.
>
> **(iii) Subgraph GNN PE similarity to [4]**: We first would like to bring to attention that the reference [4] is not a subgraph GNN PE work and is an incorrect reference for this point. Nevertheless, we explicitly acknowledge where we build our learnable GNN PE idea from (Section 3.2). Our contribution is adapting and validating this approach for relational entity graphs with stochastic initialization for improved generalization. We present this as building on prior work, not as entirely novel.
>
> Overall, we believe our work contributes significantly to learning on real-world relational entity graphs and advances the RDL sub-community in graph learning.
>
> ---
> **2. On clarifying schema defined structure**
> **Authors**: We believe our formalizations in Section 2.1 are clear. In Section 2.1, we define schema-defined structure as graphs where "edges E represent primary-foreign key relationships" between "a collection of tables T connected through inter-table relationships R" (under Definitions). This formalization distinguishes relational entity graphs from arbitrary graph structures (molecules, social networks, transportation networks, etc) where connections don't follow database schema constraints.
> Comparison with DBLP, IMDB, ACM: While these datasets share the heterogeneous property (typed nodes and edges), there are 2 critical distinctions.
> - First, temporality as a fundamental structural element: DBLP, IMDB, and ACM are static heterogeneous graphs, whereas our definition treats every entity's timestamp $\tau(v)$ as core to the graph structure, representing when entities were created/modified.
> - Second, rich multimodal entity attributes: relational databases have extensive attributes with numerical, categorical, temporal, and text fields defined by table schemas, whereas citation networks treat entities as largely symbolic with minimal attributes, such as subject area, author, paper, venue, movie genre, etc.
>
> These distinctions fundamentally change the modeling problem - existing methods for static heterogeneous graphs cannot adequately capture temporal entity evolution and attribute-rich relational patterns.
>
> ---
> **3. On claim of "RDL GNN outperform LightGBM on all tasks"**
> **Authors**: We thank the reviewer for this feedback - we have revised the phrase “across all tasks” to “across most of the tasks” in Section 2.2.
>
> ---
> (contd.)

---

> > ### Author Response · Authors · 2025-11-22
> >
> > **4. On claim of "RelGT consistently matches or outperforms GNN" and relative gains**
> > **Authors**: We would like to clarify that our claim in the abstract is supported by the results.
> > - First, across all 21 tasks, RelGT outperforms the baselines on 14 tasks, is comparable (within 1% difference) on 5 tasks, and for the remaining 2 tasks the relative gaps are 1.04% and 3.94%.
> >
> > - Second, many classification tasks could benefit from careful tuning of the multi-element tokenization components or the local and global transformer modules, as demonstrated in Table 2 and discussed in Section 4. For example, rel-avito user-clicks (a classification task predicting whether users will click on multiple ads within 4 days) shows potential for up to 8% improvement when the global module and relative time components are removed, highlighting the task-dependent nature of our modeling approach. While we do not apply task-specific tuning for every benchmark in our paper, since our primary goal is to present an effective graph transformer architecture for relational entity graphs, such tuning can be applied by practitioners to achieve additional performance gains, as discussed in Section 4.
> > - Finally, the use of relative gains follows the reporting convention in RelBench and the RDL baseline, as it provides a more consistent way to compare performance across highly diverse task types. Absolute gains may not be informative in aggregate or when compared for two different tasks (task types): for example, some regression MAE values are below 0.1 while others range up to 48, and in classification, some metrics are in the 0.9 range while others are around 0.6. We believe relative gains therefore offer a better picture when comparing two models across diverse tasks, in addition to absolute gains which are easily readable in the respective tables.
> >
> > ---
> > **5. On runtime complexity and analysis and reference [3]**
> > **Authors**: We would like to clarify that our runtime comparison in Section 4.2 shows that adding positional encodings to HGT (HGT+PE) substantially increases runtime, while not consistently improving performance compared to HGT without PE. Regarding the comparison between RelGT and the RDL baseline, we did not include a runtime analysis because it is well known that a graph transformer with all-pair attention over a set of nodes is computationally more expensive than a GNN operating on the same number of nodes, that is, O(N^2) vs. O(N). Nevertheless, graph transformers offer important advantages over GNNs, as observed in the broader graph learning literature and demonstrated in our work.
> >
> > Finally, to the best of our understanding, the method referred to as GTN in [3] is not a standard “graph transformer” as typically defined in the literature and as considered in our work (see Section 3.1). Instead, the GTN architecture applies soft-attention mechanisms, inspired by transformers, to learn and generate meta-path-based graphs from heterogeneous inputs, effectively transforming the graph structure in a differentiable, end-to-end manner. This represents a broader use of the “transformer” concept, focused on graph transformation rather than the architectural properties of graph transformers studied in our paper.
> >
> > ---
> > **6. On multiple runs**
> > **Authors**: We have included the multi-seed experimental results of RelGT in the revised manuscript (Tables 11 and 12), which are on 4 runs with different seeds. We observe that tasks with fewer test samples exhibit higher variance, consistent with the dataset statistics provided in Table 3, while tasks with larger test sets show more stable performance.
> >
> > ---
> > **7. On model complexity and scaling, and positional encoding stability**
> > **Authors**: We have revised the manuscript to include the computational complexity in Appendix A.7. Specifically, our graph tokenization fixes the local subgraph size, ensuring that the computational complexity does not increase with global graph size, and the overall scaling closely matches that of baseline methods, as discussed in our updated manuscript (Fig. 4).
> >
> > Regarding positional encoding stability, RelGT does not rely on traditional positional encodings such as Laplacian or spectral PEs. Instead, our method encodes positional information using randomized, permutation-equivariant GNN embeddings over local graph neighborhoods, in addition to type, hop, and time elements. This tokenization avoids the instability issues commonly associated with global spectral encodings, making stability analysis of those methods not directly relevant to ours.
> >
> > ---
> > (contd.)

---

> > > ### Author Response · Authors · 2025-11-22
> > >
> > > **8. On generalization to unseen tables and few-shot settings**
> > > **Authors**: While transfer to unseen tables and few-shot or foundation model scenarios are importan directions for future work, we would like to emphasize our work’s scope and current contribution. Our contribution focuses on establishing that Graph Transformers can work effectively for relational entity graphs in the standard supervised setting - a necessary foundation before exploring advanced scenarios like few-shot learning or transfer across databases. As discussed in the manuscript, naively extending existing GTs to this setting is challenging, and prior work lacks a unified architecture capable of modeling the diverse structural information in relational entity graphs.
> > >
> > > However, we believe RelGT's design is well-suited for future generalization and transfer learning scenarios. Its multi-element tokenization captures generalizable relational patterns (heterogeneity, topology, temporal dynamics) that could transfer across databases with similar schemas. Our comparison with Griffin (see Response to Reviewer 1WrR) suggests RelGT may offer advantages over GNN-based foundation models. Developing a full foundation model framework with RelGT requires addressing additional challenges including unified tokenization across database domains, pre-training task design, and scaling strategies - which we consider important future work.
> > >
> > > ---
> > > **9. On training and cost of subgraph GNN PE**
> > > **Authors**: As introduced in Section 3.1, the subgraph GNN PE is trained end-to-end with the rest of the model via backpropagation- no separate pre-training phase is required. It uses stochastic initialization where node features are randomly resampled each epoch, and the GNN layers learn to aggregate structural information from the local subgraph.
> > >
> > > Regarding computational cost, the subgraph GNN PE operates only on the local $K$-node subgraph (not the entire graph), requiring operations in the order of the edges of the subgraph. This is within the compute boundary of $O(K^2)$ of the local module of RelGT. Crucially, unlike Laplacian PE which requires expensive $O(N^3)$ global precomputation, our subgraph GNN PE requires no precomputation and is computed on-the-fly during training, making it significantly more efficient while still capturing essential structural information.
> > >
> > > ---
> > > **10. On other mentioned references [5-8]**
> > > **Authors**: We would like to refer to **point 1** and **2** of our response where we clarify the distinction between relational entity graphs and other graph types. The referenced multi-relational GNN models (except Ferrini et al., 2024) may not be directly applicable without significant architectural modifications. We note that R-GCN (Schlichtkrull et al.) does support entity classification as well as link prediction, but it is primarily developed for symbolic KGs and lacks built-in multimodal encoders or temporal mechanisms, so applying it to RelBench would require adding modality encoders and time-aware components. R-HGNN and RSHN advance relation-aware and relation-structure modeling for heterogeneous graphs but are likewise static (non-temporal) and do not directly encode rich DB schema modalities.

---

> > > > ### Comment · Reviewer_nFvD · 2025-11-26
> > > >
> > > > **1.** While you describe RELGT as the first GT for relational entity graphs, the core components are all adaptations of existing mechanisms already explored in heterogeneous, temporal, or subgraph-based Graph Transformers. Models such as HTGformer, HGT variants, and subgraph-PE Transformers already address similar combinations of heterogeneity, temporality, and structural encoding.
> > > > More generally, combining existing components can constitute a meaningful contribution, but only when supported by clear theoretical motivation explaining why their integration yields new capabilities beyond what each component already provides. In the current manuscript, this principled justification is not fully articulated, making the claimed novelty difficult to assess.
> > > >
> > > > **1 (iii).** I'm sorry for the wrong reference, my mistake.
> > > > You argue that your subgraph GNN PE is “adapted” from prior work, but this reinforces, rather than refutes, the core of my comment: the proposed PE mechanism is not novel. RELGT applies the same idea within the RDL pipeline, but the adaptation does not introduce a new conceptual contribution.
> > > > Thus, while the component may be useful, its technical novelty remains limited.
> > > >
> > > > **2.** Your rebuttal reiterates the definition from Section 2.1, but this definition (edges defined by PK–FK relationships) does not distinguish relational entity graphs from other heterogeneous graph datasets (e.g., IMDB, DBLP, ACM), which are also schema-derived with typed nodes and typed relations. Introducing dataset-level differences such as temporality or number of attributes does not justify a different architectural category.
> > > > My point remains: the notion of “schema-defined structure” as presented is not sufficiently formalized or unique to warrant a new architectural motivation.
> > > >
> > > > **4.** Regarding this point, I would like to highlight an additional fairness issue in your interpretation of the results. You state that your model “outperforms the baselines on 14 tasks and is comparable (within 1%) on 5.” However, looking at Table 1, it appears that you classify a task as “outperformed” even when RELGT wins by less than 1% in relative gain, whereas you classify a task as “comparable (within 1%)” only when RELGT loses by less than 1%.
> > > > This asymmetric use of the 1% threshold is not entirely fair:
> > > > if the 1% margin is meant to define comparability, then it should apply consistently in both directions, whether RELGT wins or loses by less than 1%.
> > > > Otherwise, the interpretation artificially inflates the number of tasks counted as “outperformed.”
> > > >
> > > > **10.**
> > > > Your argument that models like R-GCN, R-HGNN, and RSHN are not directly applicable because they lack temporal or multimodal encoders is not entirely convincing. HGT and your own model also required nontrivial adaptation to fit the RelBench setting, and many multi-relational GNNs can similarly be extended with temporal-aware sampling or multimodal encoders without altering their core functioning, in the same way that you extended Transformers.
> > > > If a model can be made compatible with RelBench through reasonable adaptations that do not fundamentally change its main mechanism, then for a benchmarking paper claiming state-of-the-art performance, those models must actually be run in order to establish a fair and complete comparison.
> > > >
> > > >
> > > > Overall, many of the issues raised above point to the same underlying concern: the paper does not position itself realistically within the existing literature, and the claimed novelty is not sufficiently supported by theoretical grounding. Strengthening these aspects would greatly improve the clarity and contribution of the work.

---

> > > > > ### Comment · Reviewer_nFvD · 2025-11-26
> > > > >
> > > > > **6.** Thank you for adding the multi-seed results. Just to fully understand the setup: were the four seeds used for Tables 11 and 12 newly executed for the revised version, or were these runs already available from the original experiments and simply not reported in the initial submission?
> > > > > Additionally, for a fully fair comparison, it would be helpful to report mean and standard deviation also for the baselines, as their performance variance is equally important for assessing the robustness of the relative differences.

---

> > > > > > ### Author Response · Authors · 2025-12-03
> > > > > >
> > > > > > We thank the reviewer for the rebuttal and the additional detailed suggestions on improving this work. We address below the key concerns, and have uploaded a revised version of the manuscript with the changes as suggested.
> > > > > >
> > > > > > **On responses to points 1,2,10: Contextualization and motivation**
> > > > > > **Authors**: In the revised manuscript, we have added Section 2.4, which clarifies the positioning of this work. This section outlines the challenges of applying existing methods, particularly graph transformers (GTs), and explains how our approach addresses them. We would like to reiterate that, although the sub-field of graph transformers has advanced in recent years for static, homogeneous graphs (and, to some extent, for temporal and heterogeneous graphs), methods such as HGT and HTGformer do not fully incorporate these developments, possibly because the problem setting spans multiple sub-fields of the graph learning literature (temporal GNNs, heterogeneous GNNs, scalability, graph transformers, relational deep learning). This gap further motivates our work, which aims to develop a graph transformer that performs competitively and often excels over the baselines in relational deep learning.
> > > > > >
> > > > > > ---
> > > > > > **On response to point 4: Interpretation of RelGT improvements**
> > > > > > **Authors**: We would like to clarify that our manuscript does not explicitly claim RelGT outperforms on 14 tasks and is comparable on 5 tasks. Our characterization in the paper (Abstract and Section 4.2) states that RelGT "consistently matches or outperforms" the baseline, with specific task-by-task results provided in Tables 1a and 1b for transparent evaluation. We used the task-count comparison (outperforms, comparable, underperforms) only in our rebuttal to address your original concerns in point 4. The <1% threshold was applied to cases where RelGT wins, naturally creating an asymmetric comparison. We appreciate your detailed observation, suggestion and have now added explicit wording in Section 4.2 for a symmetric comparison: _Using a ±1% threshold to assess comparable performance,
> > > > > > RELGT shows: (i) clear improvements (more than a 1% relative gain) on 10 tasks, (ii) comparable
> > > > > > results (within ±1%) on 9 tasks, and (iii) competitive but lower performance (more than a 1% relative
> > > > > > loss) on 2 tasks._
> > > > > >
> > > > > > ---
> > > > > > **On response to point 6**
> > > > > > **Authors**: The multiple seed results for Table 11 and 12 were available from the original experiments, as mentioned in response to reviewer ksWe. As suggested, we have revised Tables 11 and 12 to add the same for all baselines including HGT and HGT+PE which we re-ran for all 4 seeds.
> > > > > >
> > > > > > We hope this addresses your concerns.

---

### Official Review · Reviewer_Y8Ba · 2025-10-21

**Soundness:** 4
**Presentation:** 3
**Contribution:** 3
**Rating:** 6
**Confidence:** 3

**Summary:**

The paper introduces Relational Graph Transformer (RELGT), a graph transformer architecture for relational tables.
RELGT employs a multi-element tokenization strategy that includes five components (features, type, hop distance, time, and local structure). The method shows improved performance on a wide range of experiments, tested on the RelBench benchmark

**Strengths:**

1. Clear motivation, good explanation of problem

2. Architecture is well-explained, the different modules are well motivated and justified, and an ablation study analyses their benefits. Each component is linked to a specific challenge in relational deep learning, which is very good.

3. Extensive experiments: The evaluation is broad and the results are convincing. That said, as I am not an expert in all recent baselines for this specific area, I cannot definitively assess the completeness of the comparisons.

4. Good writing and clarity.

5. The work successfully leverages and extends the principles of Graph Transformers to handle the challenges of relational data.

**Weaknesses:**

### 1. Relation to Temporal Knowledge Graphs (TKG)
* There is no proper distinction between relational entity graphs and TKG (see e.g. TGB 2.0), especially in section 2.1 challenges: while I agree that relational entity graphs are difficult and challenging, and different to conventional graph data, in this section, the difference to temporal knowledge graphs is not clarified in my opinion.  in my opinion, the difference are mostly in the entity-specific attributes - do you agree? if yes,  it would be good to mention and clarify.

###  2. Integration of relation types:
* The formal definition of the REG includes relation types (psi), but these are not explicitly part of the five-element token representation. Why were edge relation types excluded from tokenization or the attention mechanism? A relational encoder or relation-aware attention might improve expressivity.

###	3. Choice of hop distance (2 hops) not motivated
* The tokenization is restricted to 2-hop neighborhoods, but this hyperparameter is not justified. Figure 4 ablates K (number of neighbors), but not the hop radius. Is RELGT sensitive to this parameter?

### 4. Figure 2 clarity:
* The left part of Figure 2 (“seed nodes with local neighbors”) is confusing to me. It is unclear how the nodes are sampled from the relational entity graph, particularly given the temporal-aware 2-hop sampling strategy, vs the fact that in each line you see 6 (plus …) nodes, where some of them are > 2 hops from the black node. A more explicit visual or step-by-step example would help readers follow the process.

### 5. Experimental infos missing
* The paper does not mention the random seed setting (fixed?) and the number of repetitions, and variance across runs.

### 6. Potential idea for improvement: Interpretability analysis
* Given that the model is relatively complex, some insights on interpretability, e.g. examples visualizing attention or token importance, would be very valuable.

## Minor:
### 7. Figure formatting
* Figures 2 and 3 have small font sizes, which makes them very hard to read.

### 8. Reproducibility, LLM usage
* The reproducibility statement and LLM usage are missing.

## Overall comment
The paper presents a valuable contribution to relational deep learning. The authors apply the concept of graph transformers, and modify and extend them to tackle the challenging task. The paper is written well, and the evaluations make sense. The ablation studies are good, the structure as well.
The paper could benefit from clarifying above questions, e.g. differentiation from TKG, improving the figure, and explaining the motivation for excluding the relation type.

**Questions:**

1. How do the challenges that you mention in section 2.1 fundamentally differ from the challenges for temporal knowledge graphs? Do you agree with my satement at W1?
2. Why were relation types excluded from the token representation?
3. How sensitive is RELGT to the 2-hops sampling assumption?
4. Were experiments repeated with fixed random seeds , and or multiple runs to estimate variance?
5. Would it be possible to integrate an interpretability analysis?

---

> ### Author Response · Authors · 2025-11-22
>
> We thank the reviewer for taking the time reading and evaluating our work, their positive feedbacks and for the suggestions on improving our manuscript. We address below the key concerns raised in the review.
>
> **1. On relation to Temporal Knowledge Graphs**
> **Authors**: We agree that relational entity graphs and temporal knowledge graphs (TKGs) share aspects of heterogeneity and temporality, but they are fundamentally different in how they represent data. TKGs (including those in TGB 2.0 which follow the standard _(subject, relation, object, timestamp)_ fact-centric formulation) model events or facts that hold at specific times, with entities treated as largely atomic symbols. In contrast, relational entity graphs are entity-centric: entities carry rich, schema-defined, multimodal attributes (numeric, categorical, text, etc.), and timestamps describe entity lifecycle changes (creation or update times) rather than fact validity. We agree with the reviewer’s statement that entity-specific attributes are a primary difference, but an equally important distinction is the presence of schema-constrained relationships (foreign-key type links) that may not have a direct analogue in TKG formulations.
>
> These differences also explain why the challenges listed in Section 2.1 can be considered differently from those in TKGs.
> - In relational entity graphs, heterogeneity arises from explicit table schemas defining distinct entity types with structured attributes, while in TKGs heterogeneity is primarily relation-type driven.
> - Temporality reflects entity-state evolution rather than temporal fact assertions, requiring reasoning over attribute trajectories rather than event sequences.
> - And schema structure introduces rigid, semantically meaningful dependencies between entity types (unlike TKGs, whose relation types are open-ended and not governed by a database schema). We will update Section 2.1 for in our final manuscript after the discussion period to clarify these differences and to explicitly highlight the reviewer’s point about entity-specific attributes.
>
> ---
> **2. On integration of relation types**
> **Authors**: While relation types are included in our formal definition of relational entity graphs, we do not introduce them as explicit token elements, since their information is already captured implicitly through other components. (i) _Node type encoding_: In relational databases, relation types are largely schema-determined by the pair of entity types involved (e.g., a “customer” linked to a “product” implies a “purchases” relation), and our node type encoder allows the Transformer to learn these schema-driven interaction patterns. (ii) _Structural encodings_: Hop distance and the subgraph GNN PE encode multi-hop relational structure, allowing sequences of relation types to be reflected in the learned PEs. (iii) _Attention mechanisms_: Prior work on heterogeneous graph transformers shows that relation-specific interaction patterns can be captured through attention even without explicit relation-type tokens, and we observe the same behavior in RelGT.
> In addition, unlike knowledge graphs with large, open-ended relation vocabularies, relational databases contain a relatively small, fixed set of relation types that are strongly governed by the schema and thus already reflected through node types and structural encodings.
>
> ---
> **3. On choice of hop distance**
> **Authors**: The choice of hops in the local neighborhood is not a restriction and can be relaxed to other numbers. We primarily use 2-hop to be consistent with the RDL baseline (Robinson et al., 2024) which also follows a 2-hop sampling for its results, thereby ensuring a fair comparison. In addition, our global module can compensate for information present in non-2-hop context.
>
> ---
> **4. On non-local nodes in local neighbor sampling**
> **Authors**: In Figure 2, the nodes shown in the “local neighbors” that appear to be more than 2 hops away occur only when the temporal-aware 2-hop neighborhood of a seed node contains fewer than K nodes. In these cases, we supplement the neighborhood by randomly sampling additional nodes from elsewhere in the graph to reach the fixed size K. This padding step ensures a consistent neighborhood size for all seed nodes.
>
> ---
> **5. On multi-seed results**
> **Authors**: We have included the multi-seed experimental results in the revised manuscript (Tables 11 and 12), which are on 4 runs with different seeds. We observe that tasks with fewer test samples exhibit higher variance, consistent with the dataset statistics provided in Table 3, while tasks with larger test sets show more stable performance.
>
> ---
> (contd.)

---

> > ### Author Response · Authors · 2025-11-22
> >
> > **6. On interpretability**
> > **Authors**: We agree that improving the interpretability of RelGT is an important direction, especially as such models are scaled up in data, pretraining tasks, and model size toward foundation models. Currently, interpretability for graph transformers remains underexplored in the literature and applying interpretability approaches, as suggested, to RelGT in the current setting may not yield reliable conclusions. For example, recent work such as [1] shows that different graph transformer variants can achieve similar performance while relying on substantially different information flows. Integrating interpretability analyses, such as visualizing attention patterns or examining token-level importance, is therefore a valuable line of future work. We view this as a promising extension of our current study.
> > [1] El, Batu, Deepro Choudhury, Pietro Liò, and Chaitanya K. Joshi. Towards mechanistic interpretability of graph transformers via attention graphs. 2025.
> >
> > ---
> > **7. On minor figure, reproducibility and LLM usage comments**
> > **Authors**: We apologize for the small font sizes in the mentioned figures-this was due to space constraints in the initial submission, and we will update the figures in the revised manuscript. We would also like to clarify that we have provided the appropriate LLM usage disclosure in the submission and included an anonymized code repository in the supplementary materials.

---

> > > ### Comment · Reviewer_Y8Ba · 2025-11-25
> > > **Reply to Official Comment by Authors**
> > >
> > > I thank the authors for taking the time to address my concerns and questions.
> > > I'll raise my score accordingly.

---

### Official Review · Reviewer_ksWe · 2025-11-01

**Soundness:** 2
**Presentation:** 3
**Contribution:** 3
**Rating:** 6
**Confidence:** 2

**Summary:**

The paper introduces Relational Graph Transformer (RelGT), a graph-transformer architecture tailored for relational deep learning (RDL), where multi-table relational databases are represented as heterogeneous temporal relational entity graphs (REGs). The core idea is a multi-element tokenization that decomposes each node into five components: node features, node type, hop distance from a seed node, relative time, and a subgraph positional encoding obtained by a lightweight GNN run on the sampled local subgraph—so that heterogeneity, temporality, and local topology are captured without heavy global precomputation. These tokens feed a hybrid local-global transformer: local attention over a sampled K-hop neighborhood plus global attention to a small set of learnable centroids updated with EMA K-means during training, yielding both neighborhood-level and database-wide context for prediction heads.

**Strengths:**

1. The multi-element tokenization is a clean, compositional alternative to “one-shot” global PEs, explicitly encoding heterogeneity, relative structure, and time, plus a lightweight subgraph PE. This is well-motivated for relational graphs and distinct from standard GTs or HGT variants.
2. Establishing a transformer baseline that consistently competes with or surpasses RDL’s hetero-GNN baseline on RelBench is meaningful; the design choices are broadly applicable in enterprise REG settings and could inform future pretraining/foundation models over relational data.

**Weaknesses:**

See Questions.

**Questions:**

1. Attribution of gains to token elements vs. architectural bias remains partially confounded. Table 2 indicates that removing subgraph PE or the global module reduces accuracy on average, but it is unclear whether RelGT’s advantage primarily comes from injecting structural bias via the GNN-based PE rather than the transformer’s attention. The fairness of comparing to HGT(+PE) without, e.g., GraphGPS-style structural encodings or stronger learned PEs on sampled subgraphs is uncertain.
2. Have you tried replacing the linear time difference with learnable spatio-temporal PEs?
3. The paper states it avoids exhaustive tuning; however, there are no multi-seed confidence intervals, paired tests, or time-to-target curves, despite notable per-task variability.

---

> ### Author Response · Authors · 2025-11-21
>
> We thank the reviewer for taking the time reading and evaluating our work, and for the suggestions on improving our manuscript. We address below the key concerns raised in the review.
>
> **1. On attribution of performance gains and HGT(+PE) comparison fairness**
> **Authors**: We would like to clarify here both the attribution of performance gains in RelGT as well comparison of HGT and HGT+PE with GraphGPS style positional and structural encodings.
>
> **i. Attribution of performance gains to token elements vs. architectural bias:** Our results suggest that both the Transformer architecture (including the global module) and the multi-element tokenization contribute to RelGT’s improvements, as in Tables 1 and 2. The ablation study in Table 2 shows that the token elements play a role, with the GNN-based PE contributing the largest performance drop when removed. Similarly, removing the global module leads to an average 3.87% accuracy decrease.
>
> As discussed in Section 3.1, standard Transformers represent tokens using two components: token features and positional structure. Graph Transformers follow the same abstraction, relying on node features and graph positional encodings (PEs). However, existing PEs are not designed to capture the unique complexities of relational entity graphs, i.e., heterogeneity, temporality, and schema-defined structures, which motivates our design. Our multi-element tokenization integrates multiple complementary aspects of relational structure that are difficult to encode within the traditional two-element framework. The ablations in Table 2 highlight that each element addresses specific modeling limitations, with certain components (e.g., GNN PE) being especially critical, while others exhibit task-dependent impact.
>
>
> **ii. Fairness of comparisons for HGT(+PE):** We believe the comparisons are favorable to HGT. HGT+PE uses Laplacian PE, which requires expensive precomputation (as shown in Figures 4 and 5), yet still underperforms RelGT on most tasks. Incorporating more structural encodings such as those used in GraphGPS is possible, but doing so would significantly increase computational costs for the HGT(+other PE) baseline while still not addressing the complexities that RelGT’s tokenization is designed for.
>
> ---
> (contd.)

---

> > ### Author Response · Authors · 2025-11-21
> >
> > **2. On replacing the linear time difference with learnable spatio-temporal PEs**
> > **Authors**: We thank the reviewer for this feedback. We would first like to clarify that RelGT's multi-element tokenization strategy, which decomposes each node into features, type, hop distance, time, and local structure is fundamentally already a spatio-temporal encoder. This design explicitly captures both spatial information (through heterogeneity via type encodings, topology via hop distance, and local structure via Subgraph GNN PE) and temporal dynamics (through relative time encodings), enabling the model to jointly incorporate the spatio-temporal information.
> >
> > However, based on the suggestion, we conducted an additional ablation study (included in Appendix A.3) exploring an explicit learnable spatio-temporal PE. Specifically, we repurposed the Subgraph GNN PE component to serve as a unified spatio-temporal encoder by initializing nodes with relative time differences $\tau(v_j) - \tau(v_i)$ between neighbors and the seed node. This approach replaces our original separate temporal encoder and Subgraph GNN PE with a single learnable module that processes temporal information through the GNN structure.
> >
> > We evaluated this unified approach across 8 diverse RelBench tasks (3 regression and 5 classification). The results in Table 5 in Appendix A.3 (also here) show that this learnable spatio-temporal PE consistently underperforms (although comparable in some cases) compared to RelGT's original multi-element tokenization design with separate encoders.
> >
> > ### Regression Tasks (MAE)
> > | Dataset | Task | Split | RelGT (Full) | RelGT (Spatio-Temporal PE) | % Rel. Diff |
> > |------------|---------------|-------|--------------|-----------------------------|--------------|
> > | rel-avito | ad-ctr | Test | **0.0345** | 0.0355 | -2.90 |
> > | | | Val | 0.0314 | 0.0315 | |
> > | rel-trial | site-success | Test | **0.3262** | 0.3554 | -8.95 |
> > | | | Val | 0.3593 | 0.3883 | |
> > | rel-hm | item-sales | Test | **0.0536** | 0.0630 | -17.54 |
> > | | | Val | 0.0627 | 0.0718 | |
> > ### Classification Tasks (AUC)
> > | Dataset | Task | Split | RelGT (Full) | RelGT (Spatio-Temporal PE) | % Rel. Diff |
> > |------------|---------------|-------|--------------|-----------------------------|--------------|
> > | rel-avito | user-clicks | Test | **0.6830** | 0.6465 | -5.34 |
> > | | | Val | 0.6649 | 0.6519 | |
> > | rel-avito | user-visits | Test | **0.6678** | 0.6641 | -0.55 |
> > | | | Val | 0.7024 | 0.7017 | |
> > | rel-event | user-ignore | Test | **0.8157** | 0.8152 | -0.06 |
> > | | | Val | 0.8868 | 0.8870 | |
> > | rel-trial | study-outcome | Test | **0.6861** | 0.6537 | -4.72 |
> > | | | Val | 0.6678 | 0.6757 | |
> > | rel-amazon | user-churn | Test | **0.7039** | 0.7036 | -0.04 |
> > | | | Val | 0.7036 | 0.7037 | |
> >
> >
> > We attribute the performance gap to **these factors**: (i) keeping temporal and spatial encodings separate allows the model to learn specialized representations for fundamentally different types of relational patterns (temporal and relational structure),  (ii) the linear time difference in our original design offers a direct signal, and (iii) the multi-element tokenization strategy's strength lies in its ability to independently encode and then combine a comprehensive view of the graph structure, which appears to be more effective than entangling spatial and temporal information within a single learnable component.
> >
> > These findings validate our design choice: while a unified learnable spatio-temporal PE is possible, RelGT's multi-element tokenization with specialized encoders for different relational aspects better captures the complex graph information w.r.t. heterogeneity, temporality, and topology in relational entity graphs.
> >
> >
> > ---
> > **3. On multi-seed variability and intervals**
> > **Authors**: We include the omitted multi-seed experimental results in the revised manuscript (Tables 11 and 12), which are on 4 runs with different seeds. We observe that tasks with fewer test samples exhibit higher variance, consistent with the dataset statistics provided in Table 3, while tasks with larger test sets show more stable performance.

---

### Official Review · Reviewer_1WrR · 2025-11-01

**Soundness:** 3
**Presentation:** 3
**Contribution:** 2
**Rating:** 6
**Confidence:** 3

**Summary:**

This work propose a new graph transformer architecture, RelGT, for RDB data.  By designing new positional encoding, add temporal information, and using structural feature extractor, RelGT achieves strong performance on RelBench.

**Strengths:**

1. Detailed ablation study verify the effectiveness of model. Tables provides ablation results of all model components on all datasets. These results show consistent performance gain of designs in this work.
2. Careful architecture design. Table 3 provides architecture illustration in detail.

**Weaknesses:**

1. Recent strong RDB baseline[1] is missing. Including it may make its contribution more clear.
2. RelGT is trained on each dataset separately, with no pretrain and transfer learning experiments, making the architectural design contribution less significant.

[1] Yanbo Wang, et al. Griffin:Towards a graph-centric relational database foundation model. ICML 2025.

**Questions:**

1. In RelGT, each node is tokenized to 5 tokens. Will it leading to significantly larger computation overhead compared with baselines?

---

> ### Author Response · Authors · 2025-11-21
>
> We thank the reviewer for taking the time reading and evaluating our work, and for the suggestions on improving our manuscript. We address below the key concerns raised in the review.
>
> **1. On Griffin as a baseline.**
>
> **Authors**: We thank the reviewer for highlighting the Griffin baseline proposed in [1]. We report here the comparison of RelGT with Griffin on 10 entity classification tasks in RelBench and have revised our manuscript with the comparison included in Appendix A.4. RelGT outperforms Griffin on 8 out of 10 tasks with relative gains of up to 8.41%, demonstrating the advantages of a Graph Transformer backbone for processing relational entity graphs, strengthening our contribution.
>
>
> | Dataset    | Task            | Griffin | RelGT (ours) | % Rel. Gain |
> |------------|------------------|---------|--------------|--------------|
> | rel-f1     | driver-dnf       | 0.745   | 0.7587       | 1.84         |
> | rel-f1     | driver-top3      | 0.825   | 0.8352       | 1.24         |
> | rel-avito  | user-clicks      | 0.630   | 0.6830       | 8.41         |
> | rel-avito  | user-visits      | 0.650   | 0.6678       | 2.74         |
> | rel-trial  | study-outcome    | 0.689   | 0.6861       | -0.42        |
> | rel-amazon | user-churn       | 0.700   | 0.7039       | 0.56         |
> | rel-amazon | item-churn       | 0.811   | 0.8255       | 1.79         |
> | rel-stack  | user-engagement  | 0.898   | 0.9053       | 0.81         |
> | rel-stack  | user-badge       | 0.870   | 0.8632       | -0.78        |
> | rel-hm     | user-churn       | 0.683   | 0.6927       | 1.42         |
>
> ---
> **2. On RelGT trained on each dataset separately, with no pretrain and transfer learning. Clarifying the scope**
>
> **Authors**: While pre-training and transfer learning are important directions for future work, we would like to emphasize our work’s scope and current contribution of establishing an effective Graph Transformer (GT) architecture specifically designed for relational entity graphs. As discussed in detail in the manuscript, naively extending existing GTs to this setting is challenging, and prior works lack a unified architecture capable of modeling the diverse structural information present in relational entity graphs.
>
> Nevertheless, we believe RelGT’s design is particularly well-suited for future pre-training and transfer-learning scenarios, and based on our comparison with Griffin in the aforementioned reply, can potentially offer advantages over GNN based designs in existing foundation models such as Griffin. Its multi-element tokenization captures generalizable relational patterns (e.g., heterogeneity, graph topology, temporal dynamics) that can transfer across relational databases with similar structural characteristics.
>
> We would also like to clarify that developing a full foundation-model framework with RelGT is part of our future work, which will require addressing additional challenges such as unified tokenization for different database domains, the design of pre-training tasks, scaling trends with model and data sizes, and fine-tuning strategies.
>
> ---
> **3. On the computation overhead of node tokenization with 5 elements**
>
> **Authors**: The proposed tokenization in RelGT does not introduce significant computational overhead. A key benefit of the multi-element tokenization is that it provides positional encoding like information without the expensive computations required in prior PE designs. We reply with the following:
>
>
> **i.** The 5 elements, features, hop distance, relative time, node type, and subgraph GNN PE, each have an encoding phase before the combined information to the Transformer network. These encoders are either lightweight models (e.g., a lightweight GNN for the GNN PE and TorchFrame for feature encoding) or simple linear projections (for the remaining three elements), as illustrated in Figure 2 of our manuscript.
> **ii.** The RDL baseline already constructs similarly sized local subgraphs using temporal-aware neighborhood sampling. In RelGT, we use a comparable number of local tokens (K = 300 in our experiments), so the number of processed nodes is similar. The main difference is that RelGT applies all-pair attention within this local context (O(K²)), whereas RDL uses linear message passing. However, this added cost is balanced by the fact that RelGT avoids the expensive Laplacian PE precomputation required in methods like HGT.
> **iii.** As shown in Figures 4 and 5, HGT with Laplacian PE incurs a 1.8x–8.62x runtime overhead over the baseline. In contrast, RelGT maintains competitive runtime while delivering better performance.
>
> Overall, all the 5 encoding components (features, type, hop, time, GNN PE) are computed independently and can be efficiently parallelized, further minimizing computational overhead.

---

### Meta-Review · Area_Chair_tTx1 · 2026-01-13

**Summary:**

This paper receives three positive scores and one negative score. Reviewer 1WrR suggests adding new baseline RDB, and the  separated training leads to the architectural design contribution less significant.  Reviewer ksWe concerns its GraphGPS-style structural encodings. Reviewer Y8Ba concerns that there is no proper distinction between relational entity graphs and TKG, and the neighborhood hyper-parameter is not justified.  Reviewer nFvD concerns that its novelty is overstated, and there are some conceptual gaps and unclear definitions.


Despite these shortcomings, Reviewer 1WrR  acknowledges its detailed ablations and careful architecture design. Reviewer ksWe acknowledges its clean multi-element tokenization. Reviewer Y8Ba acknowledges  its clear motivation and well-explained architecture. Reviewer nFvD acknowledges  its comprehensive benchmark evaluation.  The authors are encouraged to carefully polish the statement according to the suggestions of Reviewer nFvD.

**Reviewer Concerns:**

After rebuttal, a new baseline is added, the fairness of comparisons for HGT(+PE) is analyzed, the relation to temporal knowledge graphs is explored, and the model complexity and scaling are discussed. Reviewer nFvD still concerns its multi-seed experimental results.

**Reviewer Scores:**

Reviewer 1WrR may raise the score from 6 to 8  owing to the newly-added baseline and the scope explanations on separated training. Reviewer ksWe may maintain the original score, as the confounding influence of the underlying architectural bias cannot be ruled out.
Reviewer Y8Ba acknowledges  that the concerns and questions have been addressed, and accordingly will raise the score 6 to 8. Reviewer nFvD may maintain the original score, as he/she still concerns the multi-seed experiments.

---

### Decision · Program_Chairs · 2026-01-26

Accept (Poster)